# M2 cortex-dorsolateral striatum stimulation reverses motor symptoms and synaptic deficits in Huntington's disease

Sara Fernández-García[1,2,3], Sara Conde-Berriozabal[1,2,3], Esther García-García[1,2,3], Clara Gort-Paniello[1,2,3], David Bernal-Casas[4], Gerardo García-Díaz Barriga[1,2,3], Javier López-Gil[2], Emma Muñoz-Moreno[2], Guadalupe Soria[2], Leticia Campa[2,5,6], Francesc Artigas[2,5,6], Manuel José Rodríguez[1,2,3], Jordi Alberch[1,2,3,7], Mercè Masana[1,2,3]*

[1]Departament de Biomedicina, Institut de Neurociències, Facultat de Medicina i Ciències de la Salut, Universitat de Barcelona, Barcelona, Spain; [2]Institut d'Investigacions Biomèdiques August Pi i Sunyer (IDIBAPS), Barcelona, Spain; [3]Centro de Investigación Biomédica en Red sobre Enfermedades Neurodegenerativas (CIBERNED), Madrid, Spain; [4]Departament de Estadística, Facultat de Biologia, Universitat de Barcelona, Barcelona, Spain; [5]Institut d'Investigacions biomèdiques de Barcelona (IIBB), Consejo Superior de Investigaciones Científicas (CSIC), Barcelona, Spain; [6]Centro de Investigación Biomédica en Red sobre Enfermedades Mentales (CIBERSAM), Madrid, Spain; [7]Production and Validation Center of Advanced Therapies (Creatio), Faculty of Medicine and Health Science, University of Barcelona, Barcelona, Spain

*For correspondence:
mmasana@ub.edu

Competing interests: The authors declare that no competing interests exist.

**Abstract** Huntington's disease (HD) is a neurological disorder characterized by motor disturbances. HD pathology is most prominent in the striatum, the central hub of the basal ganglia. The cerebral cortex is the main striatal afferent, and progressive cortico-striatal disconnection characterizes HD. We mapped striatal network dysfunction in HD mice to ultimately modulate the activity of a specific cortico-striatal circuit to ameliorate motor symptoms and recover synaptic plasticity. Multimodal MRI in vivo indicates cortico-striatal and thalamo-striatal functional network deficits and reduced glutamate/glutamine ratio in the striatum of HD mice. Moreover, optogenetically-induced glutamate release from M2 cortex terminals in the dorsolateral striatum (DLS) was undetectable in HD mice and striatal neurons show blunted electrophysiological responses. Remarkably, repeated M2-DLS optogenetic stimulation normalized motor behavior in HD mice and evoked a sustained increase of synaptic plasticity. Overall, these results reveal that selective stimulation of the M2-DLS pathway can become an effective therapeutic strategy in HD.

## Introduction

Huntington's disease (HD) is an inherited neurodegenerative disorder with symptomatic manifestations, including involuntary movements such as chorea, dystonia, poor motor coordination, psychiatric, and cognitive symptoms. HD is caused by a CAG repeat expansion in the huntingtin (Htt) gene that translates into a polyglutamine tract in the Htt protein. Mutant Htt (mHtt) is expressed throughout the brain and disrupts a wide range of molecular pathways and signaling cascades, yet HD pathology is most prominent in the basal ganglia circuitry.

The central hub of the basal ganglia is the striatum, which controls movements and behavior through a myriad of inputs and outputs (*Calabresi et al., 2014*; *Freeze et al., 2013*; *Rothwell et al.,*

2015; Rueda-Orozco and Robbe, 2015; Tecuapetla et al., 2016). The striatum receives glutamatergic inputs from all neocortical areas and the thalamus (Bolam et al., 2000). Information flows from the cortex through the basal ganglia and it goes back to the cortex via the thalamus through two main pathways (direct and indirect), which orchestrate the proper execution of movement. Additionally, the specific origin of the cortical inputs to the striatum provides a substrate for information segregation in these circuits (Hintiryan et al., 2016; Wall et al., 2013).

In HD, motor symptoms emerge from dysregulated information flow through the basal ganglia circuits. Disturbances of the cortico-striatal communication have a leading role in HD network dysfunction, with alterations appearing in prodromal HD (Burgold et al., 2019; Dumas et al., 2013; Unschuld et al., 2012). Specifically, the caudate nucleus and the premotor cortex are predominantly affected (Unschuld et al., 2012). In HD mouse models, there is abundant evidence of a progressive disconnection between cortex and striatum (Cepeda et al., 2007; Veldman and Yang, 2018). Cortico-striatal dysfunction in HD is strongly supported by the impaired cortico-striatal-dependent motor functions (Hong et al., 2012; Puigdellívol et al., 2015), altered paired-pulse facilitation at cortico-striatal synapses (Milnerwood and Raymond, 2007), reduction of striatal excitatory postsynaptic currents with loss of cortico-striatal synapses (Cepeda et al., 2003; Deng et al., 2013); and altered glutamate release in the striatum of transgenic R6/1 mice (Nicniocaill et al., 2001). Moreover, reducing mHtt in the striatum is not sufficient to revert HD neurodegeneration, while reducing mHtt expression in both striatum and cortex ameliorates behavioral and neuropathological features of HD animals (Gu et al., 2005; Wang et al., 2014). Thus, a progressive disconnection of cortico-striatal pathways alters the information processing in basal ganglia circuitry, leading to motor disturbances. However, we do not know yet if specific cortico-striatal pathways are affected.

In this study, we aimed to further map cortico-striatal dysfunction in HD by using in vivo multimodal magnetic resonance imaging (MRI) techniques in the R6/1 HD mouse model. We took advantage of optogenetic tools (Zhang et al., 2010) and high-resolution cortico-striatal maps (Hintiryan et al., 2016) to modulate cortico-striatal function in HD mice. More specifically, we modulated the secondary motor (M2) cortex projection to dorsolateral striatum (DLS), whose structural alterations have been previously reported in HD (Hintiryan et al., 2016). Then, we used optogenetics coupled to in vivo microdialysis and to ex vivo multi-electrode arrays (MEAs) to characterize cortico-striatal dysfunction in symptomatic HD mice. With this knowledge, we aimed to test novel therapeutic interventions based on circuit restoration, an approach previously found successful for other basal ganglia disorders (Gradinaru et al., 2009; Kravitz et al., 2010).

Strikingly, selective optogenetic stimulation of the M2-DLS afferent pathway successfully rectified motor learning and coordination deficits in symptomatic HD mice. These effects were associated with improvements in synaptic plasticity such as induced long-term depression (LTD) and normalization of spine density within the striatum of HD mice. Our findings reveal that the function of the M2 cortex-DLS circuit is deeply impaired in the present HD mouse model, and indicate that selective stimulation of this pathway induces long-lasting plasticity effects that significantly ameliorate motor symptoms in HD.

## Results

### The striatum of HD mice shows decreased functional connectivity with afferent regions during rest

To evaluate distinct striatal circuit alterations in the transgenic R6/1 mouse model of HD, we studied the functional connectivity using resting-state fMRI (rs-fMRI) in ~20-week-old R6/1 mice. First, the analysis of the whole–brain network showed reduced strength (p = 0.037) and global efficiency (p = 0.029) in HD mice compared to WT mice, as assessed by the Student's t-test. In a more specific manner, we evaluated functional connectivity using seed-based analysis between the striatum and selected brain regions involved in basal ganglia function (Figure 1; Supplementary files 1–2). We found that the striatum from the left hemisphere in HD mice showed a positive correlation with a smaller brain area (252 ± 77 mm$^3$) than the corresponding in WT mice (335 ± 17 mm$^3$; p = 0.00016, Student's t-test). Then, to evaluate functional connectivity differences, we measured the mean correlation value for the regions of interest, automatically identified based on a mouse brain atlas, and analyzed genotype differences (Figure 1c). Two-way ANOVA showed a significant genotype effect

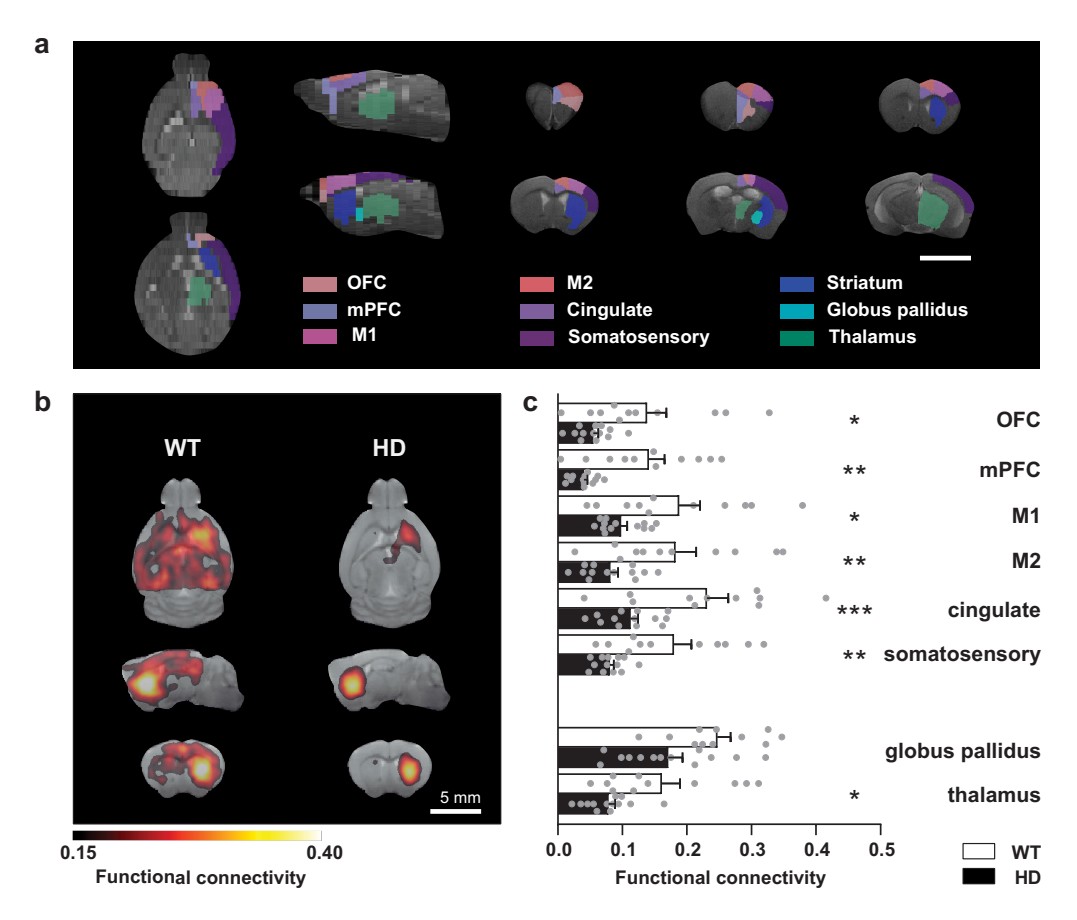

**Figure 1.** Striatal functional connectivity is reduced in symptomatic HD mice. (a) We measured the functional connectivity between selected cortical and basal ganglia-related nuclei in the regions of interest obtained by atlas-based automatic parcellation. (b) Average seed-based BOLD correlation maps from striatum in WT and the R6/1 mouse model of HD. The images show the area with an average correlation greater than 0.15. Color maps represent the average correlation value (c) Average functional connectivity of the striatum with selected cortices and basal ganglia related structures from the left hemisphere are represented. For each region, functional connectivity with striatum is computed as the average of the seed-based correlation map in the specific area. Each gray point represents data from an individual mouse. Two-way ANOVA with Bonferroni post hoc comparisons test was performed. Data are represented as mean ± SEM (WT n = 11 and HD n = 13 mice). *p < 0.05, **p < 0.01, ***p < 0.001 HD versus WT.

($F_{(1,22)}$ = 14.70; p = 0.0009) and brain region effect ($F_{(7,154)}$ = 20.67; p < 0.0001) but no region/interaction effect ($F_{(7,154)}$ = 0.6326; p = 0.7). Compared to WT, the left striatum of HD mice showed reduced functional connectivity with left orbitofrontal cortex (OFC; p = 0.0399), medial prefrontal cortex (mPFC; p = 0.0058), M2 cortex (p = 0.0055), M1 cortex (p = 0.0197), cingulate left (p = 0.006), somatosensory cortex (p = 0.007), and thalamus (p = 0.0436) but not globus pallidus (p = 0.0796), as shown by Bonferroni post hoc test. Thus, our results showed that cortico-striatal and thalamo-striatal connectivity are heavily affected in HD mice.

## Glutamate neurotransmission is reduced in the striatum of symptomatic HD mice

To further characterize cortico-striatal function in HD, we measured brain metabolites in the striatum of ~17-week-old mice, using single-voxel ${}^1$H Magnetic Resonance Spectroscopy (MRS) (*Figure 2*). Two-way ANOVA showed metabolite-genotype interaction effect ($F_{(7,154)}$ = 17.49 p < 0.0001) and Bonferroni post hoc analysis of the absolute metabolite concentrations revealed similar glutamate and glutamine, increased creatine+phosphocreatine (Cr+PCr; p = 0.0013), and decreased N-Acetyl-aspartate (NAA; p = 0.0371), NAA + NAA-Glutamate (NAAG; p = 0.0057) and Taurine (p < 0.0001; *Figure 2b*), in line with those previously described for the knock-in (KI) model of HD (*Pépin et al.,*

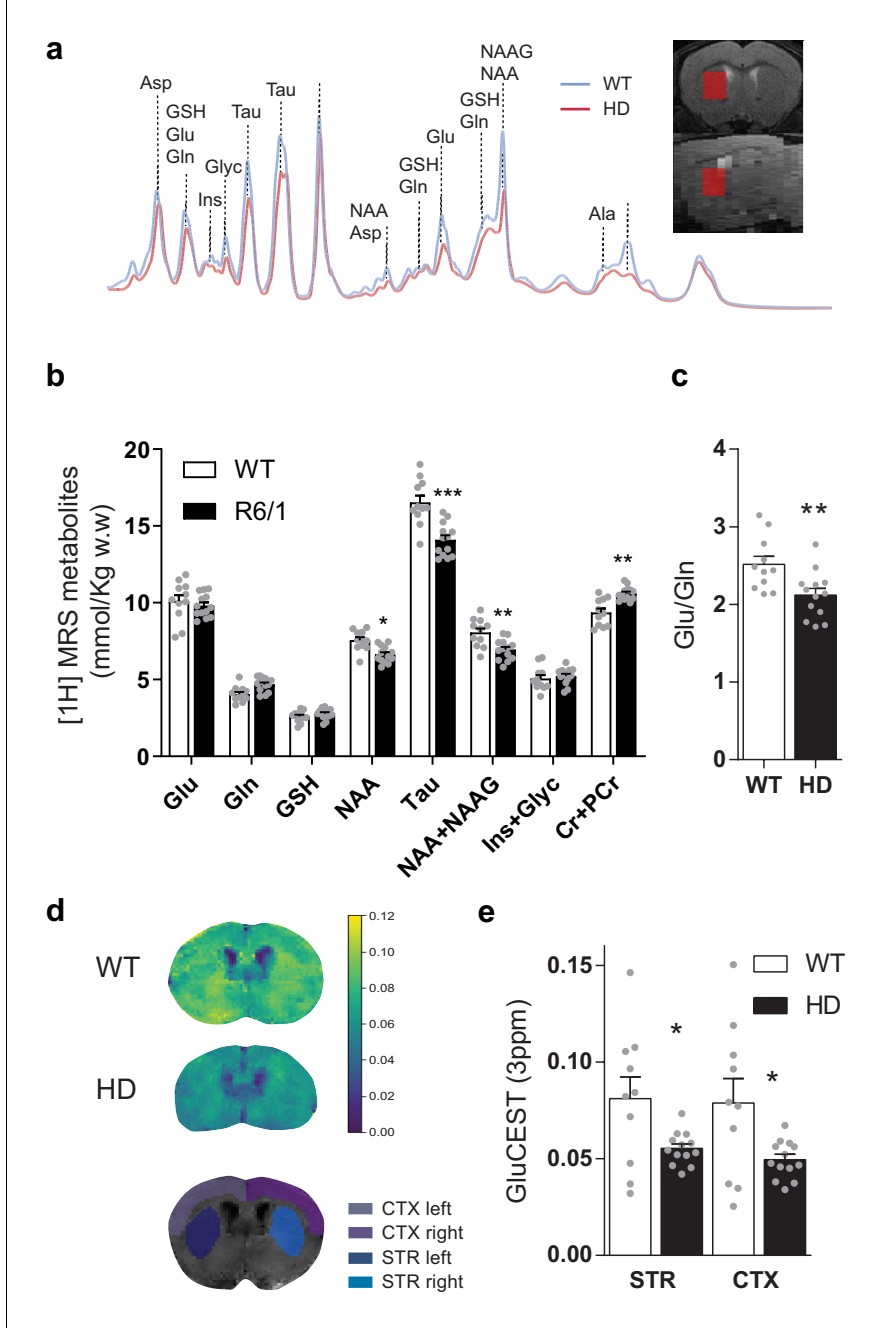

**Figure 2.** Brain metabolites in the striatum are altered in symptomatic HD mice. (**a**) Representative [1]H MRS signal from abundant metabolites in the striatum of WT (blue line) and HD (red line) mice. Voxel location in the striatum is illustrated in coronal (top panel) and sagittal brain view (bottom panel). (**b**) Metabolites concentration quantification (mmol/kg w.w.). (**c**) The glutamate/glutamine ratio was calculated as an indicator of glutamate neurotransmission (**d**) Representative coronal images of GluCEST in WT (top) and HD mice (middle) and manually drawn ROIs used for quantification (bottom). (**e**) GluCEST quantification of striatum and cortex. Values were calculated as the GluCEST value (percentage of asymmetry at 3 ppm) versus minimum GluCEST value in the ventricles and left and right hemisphere values were averaged. Data are represented as mean ± SEM (WT n = 10 and HD n = 13 mice). Each gray point represents data from an individual mouse. *p < 0.05, **p < 0.01, ***p < 0.001 HD versus WT. Abbreviations: Glu: glutamate; Gln: glutamine; GSH: glutathione; NAA: N-Acetyl-aspartate; Tau: taurine; NAAG: N-Acetyl-aspartyl-glutamate; Ins+Glyc: inositol + glycine.

The online version of this article includes the following figure supplement(s) for figure 2:

**Figure supplement 1.** Correlation of glutamate levels obtained from GluCEST and [1]H MRS in the striatum.

*2016*). Moreover, the glutamate/glutamine ratio, which is related to glutamate released as a neurotransmitter, was reduced in HD mice (p = 0.0072, Student's t-test), suggesting a reduced glutamatergic release in the striatum of HD mice (*Figure 2c*).

To study the relationship between cortical and striatal levels of glutamate, we also used chemical exchange saturation transfer (CEST) technique, which has been demonstrated to map glutamate levels (*Cai et al., 2013*; *Cai et al., 2012*) with higher sensitivity and spatial resolution compared to $^1$H MRS (*Bagga et al., 2018*; *Pépin et al., 2016*). GluCEST showed decreased glutamate levels in the striatum and cortex of HD mice compared to WT animals, as shown by two-way ANOVA brain region effect ($F_{(1,21)}$ = 16.59; p = 0.0005), genotype effect ($F_{(1,21)}$ = 6.541; p = 0.0183) but not region/genotype interaction effect ($F_{(1,21)}$ = 3.286; p = 0.0842), and Bonferroni post hoc comparisons showed significant differences between WT and HD in both regions (*Figure 2d–e*). Moreover, a significant correlation between MRS and GluCEST was observed for all cases (r = 0.5979; p = 0.0026; *Figure 2—figure supplement 1*).

## Selective M2 cortex-DLS optogenetic stimulation reveals deficient glutamate release in HD mice

To functionally assess cortico-striatal function in HD mice, we evaluated optogenetically-induced glutamate release in the striatum of WT and HD mice using in vivo microdialysis (*Figure 3a–c*). Among the different frontostriatal pathways, we chose to stimulate the projection from rostral M2 cortex to DLS based on the structural alterations previously reported in HD (*Hintiryan et al., 2016*). After six baseline dialysate samples (collected every 6 min), blue light was ON for 5 min (during sample seven collection), and six additional samples were collected in light-OFF conditions. Extracellular glutamate concentration under baseline conditions, calculated as the mean of the first six collected samples, was similar between genotypes (WT: 0.73 ± 0.08; HD: 0.69 ± 0.06 pmol/sample). Blue light stimulation led to increased glutamate levels in WT-ChR2, which returned to baseline levels after stimulation. However, increases in glutamate level were not detected in HD-ChR2 mice nor in the WT-YFP and HD-YFP control mice. Two-way ANOVA showed a significant effect of time ($F_{(12,144)}$ = 2.219; p = 0.0136), and Bonferroni post hoc analyses showed significant differences between WT-ChR2 and all other groups during stimulation (sample 7, blue box; *Figure 3b*). These data indicate a reduced cortico-striatal-dependent glutamate release in symptomatic HD mice compared to WT controls.

## Striatal neuronal response to M2 cortex afferent stimulation is dampened in HD mice

To test whether optical stimulation was effectively activating the M2 cortex-DLS pathway, we performed multi-electrode array (MEA) recordings in sagittal acute brain slices of WT-ChR2 and HD-ChR2 mice (*Figure 3d–e*). We recorded fPSC in the dorsal striatum after stimulation with increasing light intensities to create a normalized input-output assay. We found that optic stimulation of M2 cortex afferent pathways evoked fPSC in the dorsal striatum (*Figure 3e*). Two-way ANOVA showed that the amplitude of the evoked striatal fPSC was significantly larger in WT-ChR2 than in HD-ChR2 mouse (genotype effect; $F_{(10,55)}$ = 1.188; p = 0.0395) followed by Bonferroni post hoc comparisons. Also, increasing laser intensity led to fPSC of larger amplitude in both mouse groups (light intensity factor; $F_{(10,55)}$ = 9.906; p < 0.00001). These results confirm the reliability of our optogenetic approach as they provide evidence of a specific striatal response to the selective optogenetic activation of cortico-striatal afferents. Besides, the results further confirm that the M2 cortex-DLS circuit in HD is functionally altered, possibly due to both pre- and postsynaptic effects.

## Repeated M2 cortex-DLS stimulation improves exploratory and stereotypic behavior in symptomatic HD mice

Next, we examined whether the increased glutamatergic input onto DLS - induced by optogenetic stimulation of M2 cortical afferents - translated into an improved motor behavior in HD mice. To this end, we performed repeated optogenetic cortico-striatal stimulation in freely moving mice in the open field (OF) apparatus (*Figure 4*). We placed the mice in the OF and, after 5 min (PRE-stimulation), we delivered blue light for 1 min, and we left the mice in the apparatus for 5 extra min (POST-stimulation). Light stimulation was delivered at 10 Hz, which is within the physiological range of cortical pyramidal neurons firing and a similar protocol has been used for repeated cortico-striatal

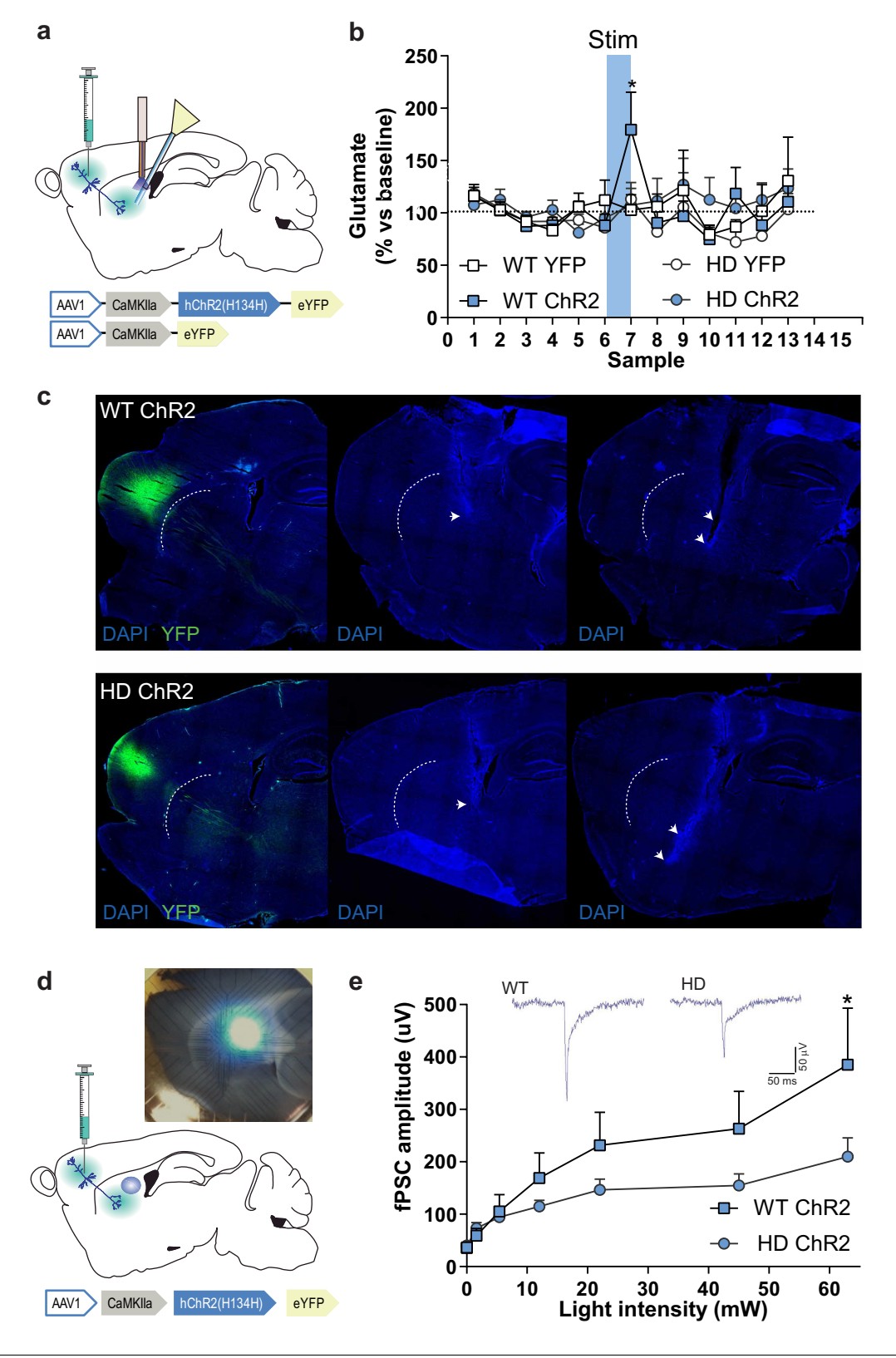

**Figure 3.** M2 cortex-dorsolateral striatum function is impaired in HD mice. (**a**) Schematic representation showing
AAV-ChR2 and control AVV-YFP constructs and injection location at M2 cortex and fiber-optic cannula and
microdialysis probe implants. (**b**) Dialysate samples were collected every 6 min and glutamate was subsequently

*Figure 3 continued on next page*

*Figure 3 continued*

analyzed by HPLC. A 473 nm light stimulation was delivered at 10 Hz bilaterally for 5 min in freely moving mice during sample 7 (blue box). Glutamate level of each dialysate sample in the striatum is represented as a percentage of the mean glutamate baseline levels (samples 1–6; WT-YFP n = 3, WT-ChR2 n = 4, HD-YFP n = 4, HD-ChR2 n = 5 mice/group). (c) Histological images from one representative WT-ChR2 (top) and one HD-ChR2 (bottom) showing AAV expressed in M2-cortex (left), fiber-optic cannula in the DLS (center), and microdialysis probe just below the fiber-optic cannula (right). Arrows indicate the tip of the fiber-optic cannula and location of the microdialysis probe membrane, respectively. (d) Schematic representation of multi-electrode array recordings in slices and a representative image showing the electrodes and the location of light stimulation. AAV-ChR2 was injected at the M2 cortex. Sagittal slices were obtained for light-induced recording. A fiber-optic cannula was used to locate the light on top of the dorsolateral striatum. (e) The amplitude of light intensity-induced striatal field postsynaptic currents (fPSC) in WT and HD mice. Inset: Representative traces of the light-induced electrical response. Two-way ANOVA with Bonferroni post hoc comparisons test was performed. Values are expressed as mean ± SEM (WT-ChR2 n = 4, HD-ChR2 n = 3 mice). *$p < 0.05$.

optogenetic stimulation (*Ahmari et al., 2013*). The distance traveled, rearing time, and stereotypies were recorded during the 11 min period (*Figure 4b and d*). We repeated the same procedure three times, each one-week apart, in symptomatic 20, 21, and 22 week-old-mice. The precision of the injection, evaluated by YFP expression, and location of fiber-optic cannula were assessed from histological slices (*Figure 4c*; *Figure 4—figure supplement 1*). While some of the neurons expressing YFP were located in M2 cortex, the presence of fluorescent fiber bundles and axons were found in cortical, striatal, and other subcortical projecting areas previously described (*Hintiryan et al., 2016*; *Reiner et al., 2010*; *Shepherd, 2013*). Fiber-optic cannulas were placed caudal in the dorsolateral striatum, avoiding direct activation of passing fibers expressing the AAV constructs.

The total distance traveled was similar in all experimental groups during OF sessions, and light stimulation did not induce significant changes in locomotor activity before, during, or after the light stimulation (*Figure 4e*, *Figure 4—figure supplements 2a* and *3a*). Two-way ANOVA with group and session as factors was performed before and after optogenetic stimulation (*Figure 4e*). The distance traveled did not show significant interaction ($F_{(6,104)} = 0.4931$; p = 0.8123), time ($F_{(2,104)} = 1.143$; p = 0.3227) or group effects ($F_{(3, 104)} = 1.415$; p = 0.2425) during the 5 min PRE period, while showed group effects during 5 min POST stimulation ($F_{(3,104)} = 3.856$; p = 0.0116) but not interaction ($F_{(6,104)} = 0.3272$; p = 0.9214) or time ($F_{(2,104)} = 0.4005$; p = 0.6710) effects. Bonferroni post hoc test failed to show significant main group effects, although there was a tendency for HD-ChR2 to increase the distance traveled compared to HD-YFP after stimulation.

By contrast, exploratory behavior, measured as rearing time, was reduced in HD mice compared to WT, as previously described for KI mice (*Pépin et al., 2016*). Optogenetic stimulation induced an increase of exploratory behavior over the sessions in HD-ChR2 mice (*Figure 4f*, *Figure 4—figure supplements 2b* and *3b*). Two-way ANOVA with group and session as factors was performed before and after optogenetic stimulation (*Figure 4f*). The rearing time showed significant group effect ($F_{(3,104)} = 6.777$; p = 0.0003) but not interaction ($F_{(6,104)} = 1.174$; p = 0.3261) or time ($F_{(2,104)} = 1.199$; p = 0.3057) effects during the 5 min PRE period; and showed group and time effects during 5 min POST stimulation ($F_{(3,104)} = 9.199$; p < 0.0001) and ($F_{(2,104)} = 3.472$; p = 0.0347, respectively) but not interaction ($F_{(6,104)} = 0.6651$; p = 0.6780). Bonferroni's main group comparison revealed that the HD-YFP group significantly showed reduced exploratory behavior compared to all other groups before stimulation, indicating that exploratory behavior in stimulated HD mice increased over the sessions and that the effects last for at least 1 week. Moreover, after the stimulation WT-ChR2 group showed reduced rearing time compared to WT-YFP, and it was no longer different from HD-YFP and HD-ChR2 groups, indicating an acute effect of the optogenetic stimulation in WT mice. Analysis of exploratory behavior during the 11 min period (*Figure 4—figure supplements 2b* and *3b*) showed that exploratory behavior in HD mice is increased over the sessions, further indicating long-lasting effects of the optogenetic stimulation on rearing time.

Also, we measured stereotypic grooming in the same mice (*Figure 4g*, *Figure 4—figure supplements 2c* and *3c*), as repetitive grooming is associated with stereotypic pathological behavior that may reflect cortico-striatal dysfunction. In non-stimulated conditions, HD mice showed an increased stereotypic behavior, as compared to WT mice, in line with the previously described increase in

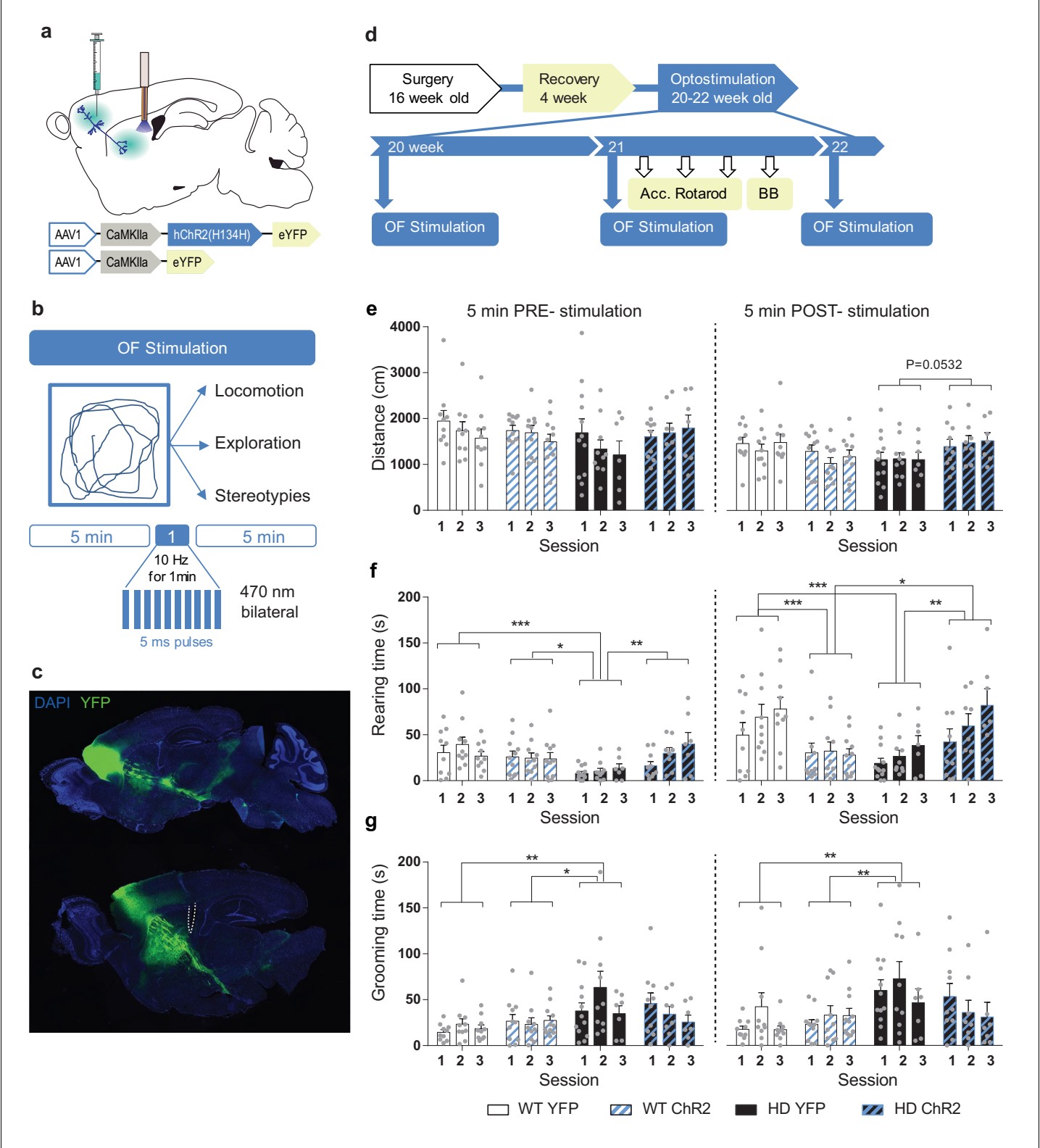

**Figure 4.** Repeated optogenetic stimulation of the cortico-striatal pathway recovers exploratory and stereotypic behavioral deficits in symptomatic HD mice. (a) Schematic diagram showing AAV-ChR2 and control AAV-YFP constructs injections at the M2 cortex and fiber-optic cannula implant in the DLS allowing light stimulation of cortically infected neuronal axons. (b) Locomotion (distance traveled), exploration time (rearing time), and stereotypic grooming (grooming time) were evaluated during 11 min open field session. Optogenetic stimulation consisted of 10 Hz light stimulation for 1 min
*Figure 4 continued on next page*

Figure 4 continued

during open field (OF) task. (c) Representative fluorescence image showing the precision of the AAV injection by YFP expression at M2 cortex and the presence of YFP fibers in the striatal region. (d) Surgery, behavior, and optogenetic experimental timeline. The OF procedure was performed at 20 (1st day OF), 21 (2nd day OF), and 22 (3rd day OF) weeks old mice. After 2nd stimulation day, motor learning and coordination tests were performed. (e) Distance traveled, (f) rearing time, and (g) stereotypic grooming were measured during the 5 min before (PRE-stimulation) and the 5 min after (POST-stimulation) over the three OF sessions, left and right panels, respectively. Values are expressed as mean ± SEM (WT-YFP n = 10, WT-ChR2 n = 11, HD-YFP n = 13, and HD-ChR2 n = 11). Each point represents data from an individual mouse. Data were analyzed by two-way ANOVA with genotype and light stimulation as factors, and Bonferroni test as a post hoc. *p < 0.05, **p < 0.01, and ***p < 0.001.

The online version of this article includes the following figure supplement(s) for figure 4:

**Figure supplement 1.** Validation of the AAV expression from the M2 cortex and its projections by YFP fluorescence.
**Figure supplement 2.** Average locomotion, exploration and stereotypies over the 11 min OF sessions.
**Figure supplement 3.** Representation of the changes in locomotion, exploration and stereotypies per minute during the whole 11 min OF sessions.

grooming in KI mice (*Pépin et al., 2016*). Interestingly, repeated optogenetic stimulation reduced stereotypical behavior in HD-ChR2 mice. Two-way ANOVA with group and session as factors showed significant group effect ($F_{(3,104)}$ = 5.073; p = 0.0026) but not interaction ($F_{(6,104)}$ = 1.200; p = 0.3124) or time ($F_{(2,104)}$ = 1.029; p = 0.3609) effects during the 5 min PRE period; and also showed significant group effects ($F_{(3,104)}$ = 4.931; p = 0.003) but not interaction ($F_{(6,104)}$ = 0.8105; p = 0.5641) or time ($F_{(2,104)}$ = 1.315; p = 0.2729) effects during the 5 min POST period. Bonferroni's main group comparison revealed that the HD-YFP group significantly showed increased stereotypic grooming before and after stimulation. Moreover, analysis of stereotypic behavior during the whole 11 min period and for each min showed a significant reduction of stereotypic grooming during the last OF session (*Figure 4—figure supplements 2c* and *3c*).

Overall, our data revealed that M2 cortex-DLS stimulation had a stronger impact on the modulation of spontaneous activity of HD mice, while acute locomotor activation was not observed. The effects of the light stimulation on exploratory behavior were manifested before and after stimulation over the sessions, suggesting some long-lasting effects evoked by the optogenetic stimulation of this M2-DLS circuit (*Figure 4—figure supplement 3c*).

## Motor learning and coordination deficits in symptomatic HD mice are recovered by optogenetic stimulation of the M2-DLS cortico-striatal pathway

We further explored the effects of M2-DLS stimulation on motor learning and coordination. Motor learning was assessed by measuring the latency to fall from an accelerating rotarod. HD mice showed reduced latency to fall compared to WT, as previously described (*Creus-Muncunill et al., 2019*; *Puigdellívol et al., 2015*). Interestingly, optogenetic stimulation of the M2 cortex-DLS pathway increased the latency to fall from the rod in HD-ChR2 mice to similar levels as WT mice. Repeated measures ANOVA showed significant effects of mice group ($F_{(3,35)}$ = 6.058; p < 0.0020), time ($F_{(11,385)}$ = 24.24; p < 0.0001), and interaction ($F_{(33,385)}$ = 2.160; p < 0.0003). Moreover, Bonferroni post hoc analyses showed significant differences between WT-YFP and HD-YFP, which are mostly lost in HD-ChR2 (*Figure 5c*).

Besides, we evaluated motor balance and coordination by measuring the ability of mice to walk across a balance beam for 2 min (*Figure 5d*). The number of 5 cm-frames crossed was reduced in HD-YFP mice compared to WT mice, as already described (*Creus-Muncunill et al., 2019*). Remarkably, M2 cortex-DLS stimulation ameliorated motor coordination in HD-ChR2 mice, reaching similar levels to WT mice. Two-way ANOVA showed a genotype effect ($F_{(1,34)}$ = 7.226; p < 0.011), with no stimulation ($F_{(1,34)}$ = 1.270; p = 0.267) or interaction ($F_{(1,34)}$ = 2.863; p = 0.0998) effects, and Bonferroni post hoc analyses showed significant differences only between WT-YFP and HD-YFP mice groups. These data suggest that the prior activation of the M2-DLS circuit induces long-lasting effects that impact the basal ganglia motor function. Moreover, our results indicate that by repeated activation of selected cortico-striatal circuits, that is, M2-DLS circuit, we can successfully ameliorate motor function in HD.

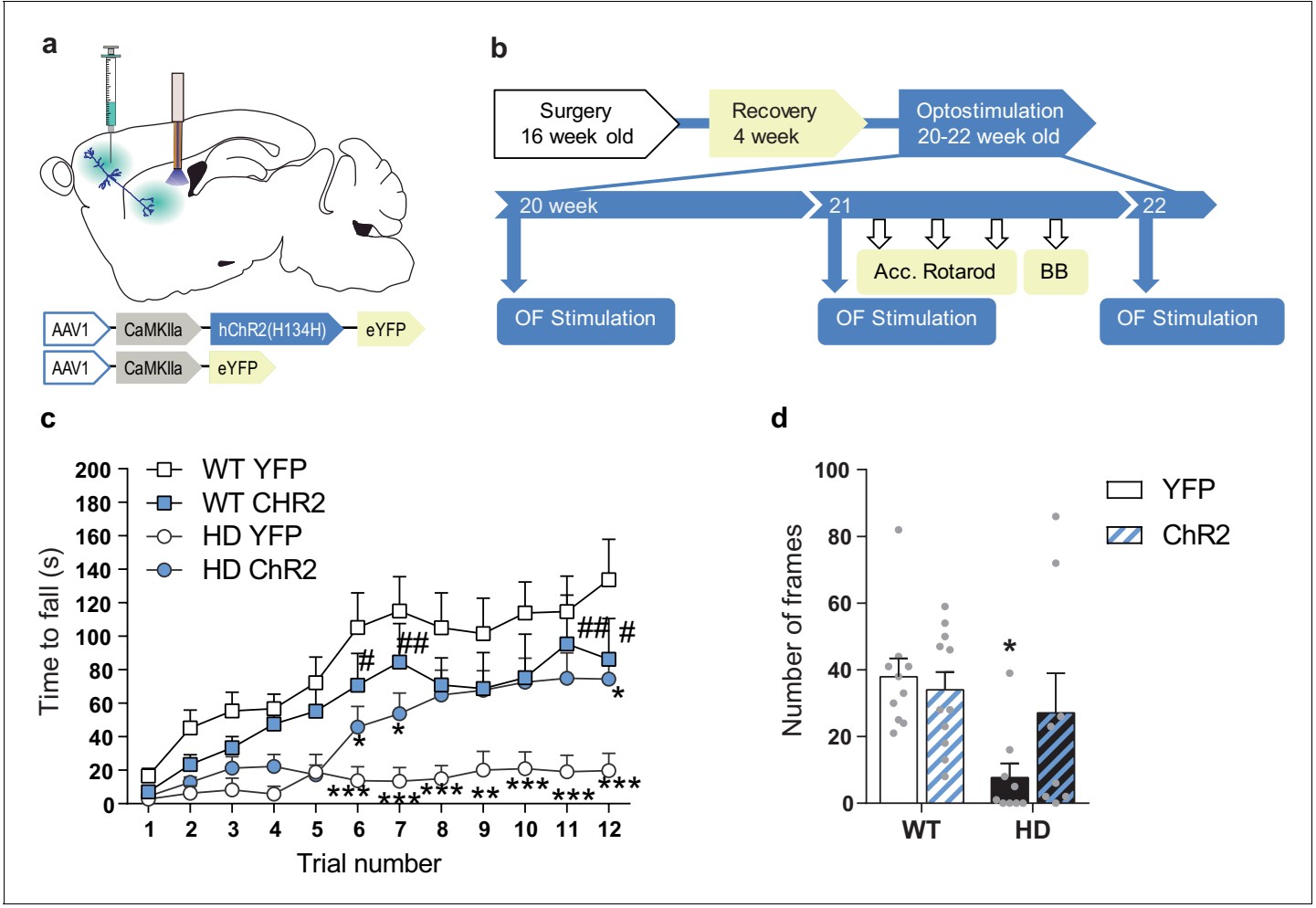

**Figure 5.** Optogenetic stimulation of the M2 cortex-DLS pathway recovers motor learning and coordination deficits in HD mice. (a) Schematic representation showing AAV-ChR2 and control AVV-YFP constructs injections at the M2 cortex, fiber-optic cannula implant in the DLS. (b) Surgery, behavior, and optogenetic experimental timeline. After 2nd stimulation day in the OF, motor learning (accelerating rotarod) and coordination (balance beam, BB) tests were performed. (c) Latency to fall in the accelerating rotarod task. (d) The number of frames crossed in the BB test. Each point represents data from an individual mouse. Data were analyzed by repeated-measures ANOVA with group and time as factors for accelerating rotarod and by two-way ANOVA with genotype and stimulation as factors for the BB test. Bonferroni's multiple comparison test was performed as a post hoc test *p < 0.05, **p < 0.01, ***p < 0.001 versus WT-YFP, and #p < 0.05 versus HD-YFP. Values are expressed as mean ± SEM (WT-YFP n = 10, WT-ChR2 n = 11, HD-YFP n = 10, and HD-ChR2 n = 9).

## Repeated cortico-striatal stimulation triggers persistent improvements of synaptic plasticity in symptomatic HD mice

We explored whether these behavioral effects were due to the long-lasting synaptic re-wiring of the cortico-striatal circuit in HD (*Figure 6*). We repeated the M2-cortex-DLS stimulation procedure in vivo (*Figure 6—figure supplement 1*) and then explored if the repeated optogenetic stimulation modulates the electrophysiological response in cortico-striatal sagittal slices. On the 3rd stimulation week, we obtained cortico-striatal slices and recorded fPSC response to increasing light intensity stimulation. We found that increasing laser intensity led to striatal fPSC of increasing amplitude in both WT-ChR2 and HD-ChR2 groups as shown by two-way ANOVA light intensity effect ($F_{(7,154)}$ = 39.05; p < 0.0001). Interestingly, we found no genotype effect or genotype/light intensity interaction (*Figure 6c*), suggesting that repeated light stimulation could be involved in these effects.

Then, we recorded fPSC in the striatum after the electrical stimulation of cortical afferents. We induced LTD, which is known to be altered in HD mouse (*Creus-Muncunill et al., 2019*; *Cummings et al., 2006*; *Ghiglieri et al., 2019*; *Li et al., 2015*), using a theta-burst stimulation (TBS)

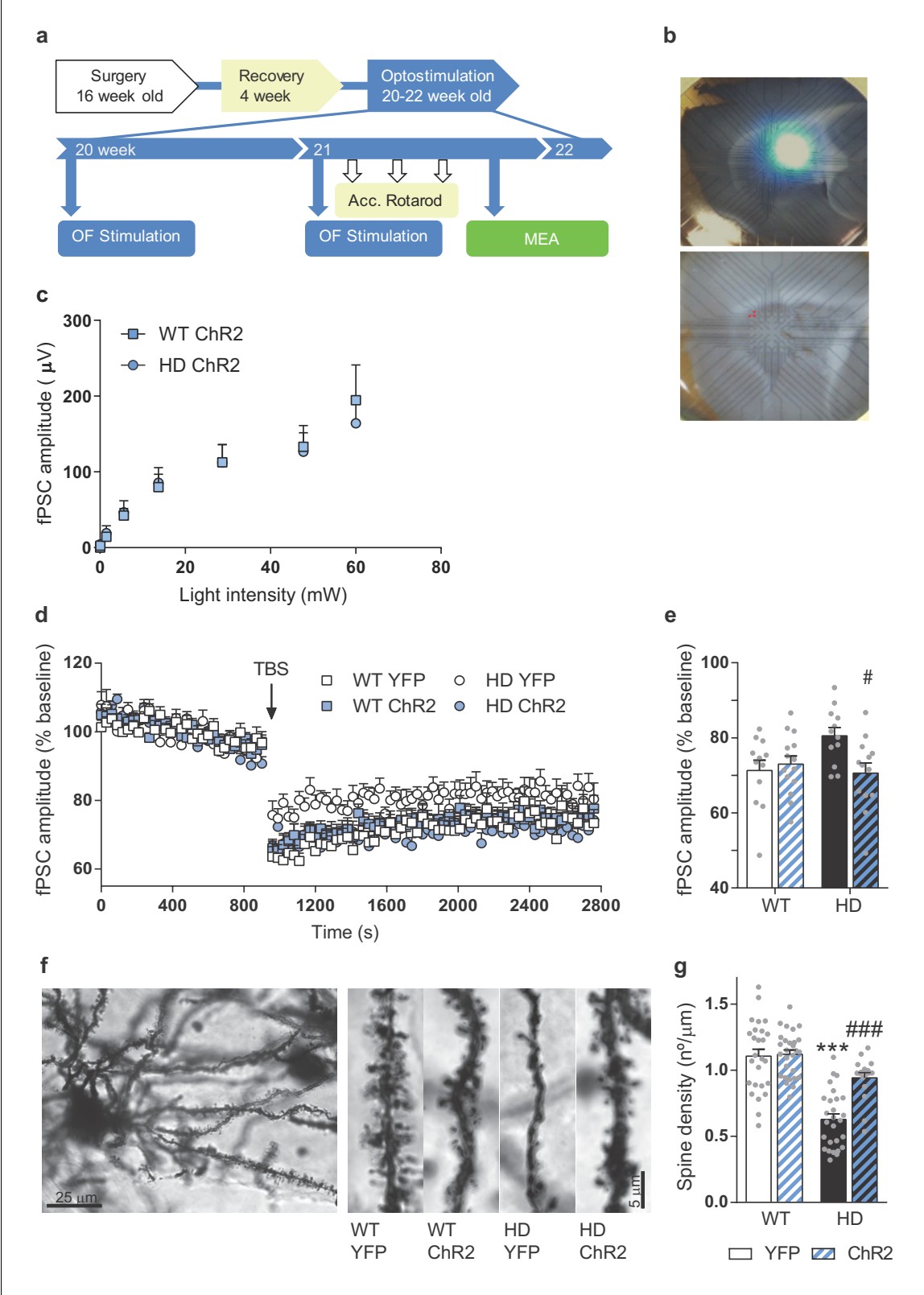

**Figure 6.** Repeated optogenetic cortico-striatal stimulation restores synaptic plasticity and dendritic spine density in HD mice. (a) Surgery, behavior, and optogenetic experimental timeline. The electrophysiological evaluation was performed on the 3rd week in sagittal cortico-striatal slices. (b) A representative image showing the location of the light stimulation (top image) and stimulating electrodes (bottom image, highlighted in red). (c) Light intensity-induced striatal field postsynaptic currents in WT and HD mice with AAV-ChR2 expressed in M2 cortex. (d) The graph shows the time course of

*Figure 6 continued on next page*

*Figure 6 continued*

LTD evoked at cortico-striatal synapses following a TBS. The TBS, indicated by the arrow, was presented after 15 min of baseline recordings. Field postsynaptic currents (fPSC) are represented as a percentage of baseline. (e) The histogram shows the averaged amplitude of fPSC evoked during 30 min after TBS. Data were analyzed by repeated-measures ANOVA with genotype and stimulation as factors and Bonferroni's multiple comparison as post hoc test. Values are expressed as mean ± SEM (WT-YFP n = 12, WT-ChR2 n = 13, HD-YFP n = 12, and HD-ChR2 n = 14 mice). (f) Golgi-impregnated representative neuron and segments of secondary dendrites from MSNs from WT and HD mice infected with AAV-YFP and AAV-ChR2, respectively. Scale bar, 3 μm. Dendritic spines were counted in a segment of known length (~20 μm) to obtain the spine density. (g) Quantitative analysis of spine densities per μm of the dendritic length of 3–13 neurons per mice. HD-YFP mice exhibit a significant reduction in dendritic spines that was significantly increased in stimulated HD-ChR2. Two-way ANOVA with Bonferroni's post hoc comparisons test was performed. All data are shown as the mean ± SEM. ***$p < 0.001$ versus WT-YFP and #$p < 0.05$ and ###$p < 0.001$ versus HD-YFP.

The online version of this article includes the following figure supplement(s) for figure 6:

**Figure supplement 1.** Behavioral data generated in mice used for *Figure 6a–e*.

protocol on cortical afferents and compared the amplitude of evoked striatal fPSC (*Figure 6d–e*). Both, WT-YFP and WT-ChR2 mice, presented a significant decrease in the amplitude of evoked fPSC following the LTD protocol that lasted at least 30 min, indicating cortico-striatal LTD induction. LTD was also slightly induced in HD-YFP, but to a less extent than WT-YFP (Student's t-test, p = 0.0150). In the same conditions, stable LTD was induced in HD-ChR2 animals to the same extent as WT mice. Two-way ANOVA showed a mice group/time interaction ($F_{(273, 4277)} = 1.573$; $p < 0.0001$), time effect ($F_{(91, 4277)} = 111.9$; $p < 0.0001$) and group effect ($F_{(3,47)} = 3.312$; $p = 0.0279$) and main group effects showed differences between HD-YFP and HD-ChR2 (p = 0.04), as shown by Bonferroni post hoc test. Indeed, similar results were obtained when analyzing the average of the fPSC amplitude after TBS (*Figure 6e*). Here, two-way ANOVA showed a genotype/stimulation interaction ($F_{(1,47)} = 5.621$; $p = 0.0219$), and Bonferroni's post hoc test showed that optogenetic stimulation restored the ability to induce LTD in HD mice.

To examine whether improved cortico-striatal synaptic function correlated with plasticity changes, we evaluated spine density from Golgi-impregnated striatal MSNs (*Figure 6f–g*). Spine density in striatal MSNs from HD mice was reduced compared to WT mice, as previously shown for MSNs from 3-month-old YAC128 (*Marco et al., 2013*) and for hippocampal apical dendrites from R6/1 (*Miguez et al., 2015*). Repeated optogenetic stimulation increased spine density in HD mice, without altering spine density in WT mice, as indicated by two-way ANOVA with genotype ($F_{(1,94)} = 56.52$; $p < 0.001$), stimulation ($F_{(1,94)} = 14.29$; $p < 0.0003$) and interaction effect ($F_{(1,94)} = 11.77$; $p < 0.0009$). Bonferroni post hoc comparisons showed significant differences between WT-YFP and HD-YFP (p < 0.0001), and between HD-YFP and HD-ChR2 (p < 0.0001), indicating that repeated stimulation of the cortico-striatal pathway restored spine density in the striatum of HD mice.

Altogether, our results show that optogenetic M2-striatal stimulation induces persistent synaptic plasticity changes that could explain the behavioral improvements found in HD mice.

## Discussion

Basal ganglia circuit dysfunction is the leading cause of motor abnormalities in HD. In particular, cortico-striatal disconnection has a prominent role in disease progression. Here we further characterized cortico-striatal functional alterations in the symptomatic HD mouse. Also, our data highlight that the M2-DLS pathway is profoundly compromised in our R6/1 mouse model of HD. Modulation of brain circuits using optogenetic tools has emerged as a promising therapeutic strategy in many neurological disorders, such as Parkinson (*Gradinaru et al., 2009*; *Kravitz et al., 2010*; *Magno et al., 2019*), stroke (*Cheng et al., 2014*; *Tennant et al., 2017*), and psychiatric disorders (*Fuchikami et al., 2015*). In this line, we aimed to restore circuit function in HD by increasing selected cortico-striatal activity in symptomatic mice. Indeed, we provide strong evidence that we can ameliorate HD symptoms in symptomatic HD mice by stimulating the M2 cortex-DLS pathway.

## Cortico-striatal functional connectivity and glutamate transmission are reduced in HD mice

Our results indicate general cortico-striatal and thalamo-striatal functional connectivity deficits in HD mice, which are in line with previously described alterations in striatal inputs (*Reiner and Deng, 2018*). However, synaptic loss, as well as electrophysiological responses in cortico-striatal or tha-lamo-striatal circuits, are differentially affected in HD (*Deng et al., 2013*; *Kolodziejczyk and Raymond, 2016*; *Parievsky et al., 2017*; *Reiner and Deng, 2018*). Also, topographically selective changes in the cortex explain the clinical heterogeneity found in HD. For example, larger primary motor cortex degeneration correlates with motor impairments while cingulate cortex degeneration correlates with predominant mood symptomatology (*Estrada-Sánchez and Rebec, 2013*; *Rosas et al., 2008*; *Thu et al., 2010*). Therefore, further understanding how specific inputs to the striatum are altered could help to better understand the progression of HD symptoms.

Indeed, the cortex comprises the primary source of striatal glutamate, however, whether its total levels are altered in HD remains unclear. Cortical and striatal metabolites, measured by [1]H MRS, are known to be profoundly altered in symptomatic HD models (*Zacharoff et al., 2012*), and glutamate concentration, measured by GluCEST, is known to be reduced in different brain regions, including most of the basal ganglia nuclei involved in motor behavior (*Pépin et al., 2016*). In our R6/1 mouse model, we also obtained divergent results. Brain metabolites, measured using [1]H MRS, showed significant changes in line to the ones observed in KI (*Pépin et al., 2016*), R6/2 (*Tkac et al., 2007*), and in vitro MRS striatal extracts from R6/2 mice (*Jenkins et al., 2000*). However, when evaluating absolute glutamate values by [1]H MRS as well as baseline glutamate levels using microdialysis, we failed to show total glutamate alterations in the striatum of HD mice, in line with the unaltered glutamate levels described in R6/2 mice (*Zacharoff et al., 2012*). Conversely, GluCEST values indicate reduced glutamate concentration in both striatum and cortical regions. While further studies are needed to clarify alterations of total glutamate levels (which includes non-synaptic sources [*Ramadan et al., 2013*; *van der Zeyden et al., 2008*]); the glutamate/glutamine ratio (whose value is linked to functional/released synaptic glutamate) is consistently reduced in the striatum of HD mice, indicating that glutamatergic neurotransmission is defective in HD.

We then evaluated glutamate release from cortical terminals. Optogenetic stimulation of cortical afferents in the striatum induced a small glutamate increase in WT mice but failed to detect increases in glutamate (over baseline levels) in HD mice, indicating deficits in glutamate release in symptomatic stages. This effect is further supported by the blunted electrophysiological response to M2 optogenetic acute stimulation observed in the striatum of HD mice. Although previous work described higher glutamate release in R6/1 mice (*Nicniocaill et al., 2001*), these discrepancies could be explained by the different location of the microdialysis probe (dorsomedial striatum [DMS] versus DLS), mice age (16 versus 20 week-old-mice), and strategy to induce glutamate release (application of KCl at high concentration or NMDA versus selective optogenetic stimulation), which makes challenging to evaluate and compare sources of the glutamate measured (cortico-striatal versus thalamic or even astrocytic).

Moreover, convergent evidence indicates impaired glutamate release in HD, although most of them rely on electrophysiological measurements (*Estrada-Sánchez and Rebec, 2013*; *Klapstein et al., 2001*; *Milnerwood and Raymond, 2007*). Indeed, the monitoring of synaptic changes during the progression of HD in the YAC128 model reported biphasic age-dependent alterations of the cortico-striatal activity (*Joshi et al., 2009*). Asymptomatic HD mice present increased glutamate release, postsynaptic currents, and synaptic responses. However, there is a shift in the three parameters as symptoms appear, which are maintained at late stages. Thus, symptomatic YAC128 mice exhibit reduced glutamate release, postsynaptic currents, and synaptic responses. Altogether, these data indicate that reduced glutamate release from the cortex, at least in late stages of the disease, might lead the basal ganglia network alterations found in HD patients.

## Motor alterations in HD involve M2 cortex-DLS cortico-striatal dysfunction

Motor learning and coordination have been extensively characterized in animal models of HD (*Creus-Muncunill et al., 2019*; *Hong et al., 2012*; *Puigdellívol et al., 2015*). In this work, we support the idea that M2-DLS activity is necessary for the proper execution of motor learning and

coordination tasks, as by manipulating this specific cortico-striatal subcircuit we restore motor defects in pathological HD mice. Indeed, M2 lesions impair acquisition and reversal of a simple sequence consisting of two lever presses, one distal and the other proximal to reward (*Yin, 2009*). Also, there is an extensive functional reorganization in the striatum during different phases of learning during a motor learning task, such as accelerated rotarod (*Yin, 2009*) or serial order task (*Rothwell et al., 2015*). This includes initial predominant activity in DMS during early training, while extensive training engages DLS, indicating that DLS has a crucial role in the completion of a learned sequence (*Rothwell et al., 2015*; *Yin, 2009*). Moreover, the activity of MSNs from direct or indirect output pathways in the DLS also experiences divergent firing patterns during a motor learning task (*Jin et al., 2014*). Action initiation correlates to increased activity in both dMSNs and iMSNs in DLS, and, as the action sequence progress, most iMSNs reduce firing rate while dMSNs exhibit sustained activity (*Jin et al., 2014*; *Rothwell et al., 2015*). Taking this together, we suggest that our M2-DLS optogenetic stimulation might strengthen the ability to complete a motor learning task in HD mice.

In this line, the proportion of cortico-striatal inputs to the direct or indirect pathways depends on the cortical subregion from where these inputs arise (*Wall et al., 2013*). For example, the primary motor cortex sends axonal projections to a greater extent to the indirect pathway, while the sensory cortex projects more to the direct pathway, and the secondary motor cortex projects similarly to both striatal output circuits (*Wall et al., 2013*). Besides, cortical sparse long-range GABAergic projection neurons also modulate striatal output and motor behavior (*Melzer et al., 2017*; *Rock et al., 2016*). Specifically, somatostatin, but not PV, GABAergic neurons from M2 inhibit movement by both D1 and D2 MSNs inhibition, while somatostatin GABA long-range projections from M1 increase movement by inhibiting cholinergic interneurons in the striatum (*Melzer et al., 2017*). Thus, there is an urgent need to precisely map region and cell-specific striatal inputs to understand the functional organization of the cortico-striatal subcircuits and their impact on behavior, both under physiological and pathological conditions.

## Repeated M2 cortex-DLS stimulation leads to sustained synaptic plasticity changes and rectifies motor deficits in HD

Our data demonstrate that selective optogenetic stimulations induce long-lasting effects in neuronal plasticity and behavior, which are fundamental for brain rewiring in pathological conditions. Optogenetic stimulation of the infralimbic prefrontal cortex induces sustained antidepressant behavioral and synaptic effects (i.e. spine density; *Fuchikami et al., 2015*), and optogenetic stimulation of the motor cortex promotes functional recovery after stroke (*Cheng et al., 2014*). Also, repeated cortico-striatal optogenetic stimulation generates persistent stereotypic behavior in mice (*Ahmari et al., 2013*) that lasts at least 2 weeks, which is specifically dependent on the cortico-striatal subcircuit stimulated (i.e. orbitofrontal cortex [OFC]–ventromedial striatum [VMS]; *Ahmari et al., 2013*). In contrast to our results, OFC–VMS stimulation induces grooming behavior in mice (*Ahmari et al., 2013*), pointing out that the specific effects are dependent on the distinct cortico-striatal subcircuits.

More surprising was the finding that M2-DLS stimulation was able to ameliorate exploratory behavior and stereotypies in HD mice. Our data show a decrease in rearing and an increase in stereotypic grooming time in our R6/1 mice compared to WT, as was previously observed in KI mice (*Hong et al., 2012*; *Pépin et al., 2016*). Also, we were able to induce exploratory activity, measured as rearing time, and reduce stereotypic grooming in already symptomatic mice by M2-DLS optogenetic stimulation. Active exploration and grooming modulate cortico-striatal phase synchrony in R6/2 mice (*Hong et al., 2012*), indicating that cortico-striatal function is involved in these behaviors. Here we further demonstrate that these spontaneous behaviors can be modulated by the M2-DLS pathway, and again, the effects are persistent at least 1 week, as changes are already observed before the light stimulation over the sessions. Therefore, optogenetic manipulations at cortical circuits generate sustained synaptic plasticity changes. However, if the long-lasting effects are specific of cortical circuits or are a general feature of optogenetic stimulation remains to be determined.

Overall, we highlight cortico-striatal circuits to be deeply altered in symptomatic HD mice. Moreover, by taking advantage of optogenetic techniques, we demonstrated altered glutamate release from M2 cortical projection neurons and deficient response in the striatum in HD. In future studies, the ability to dissect the function of different cortico-striatal subcircuits in physiological and pathological conditions might help to further understand how mHtt alters the information processing in the basal ganglia. Here, we highlight a key role of the M2 cortex-DLS specific subcircuit in the

modulation of motor alterations in HD, including exploratory behavior, stereotypies, motor learning, and coordination. In this context, we were able to successfully reestablish physiological, morphological, and behavioral alterations in our symptomatic HD mice by generating persistent synaptic plasticity in selected cortico-striatal circuits. Our successful approach opens new opportunities to design therapeutic strategies to ameliorate HD symptoms based on circuit restoration, which might be useful also for other basal ganglia disorders.

# Materials and methods

## Key resources table

| Reagent type (species) or resource | Designation | Source or reference | Identifiers | Additional information |
|---|---|---|---|---|
| Genetic reagent *Mus musculus* | B6CBA-Tg(HDexon1) 61Gpb/1J (R6/1) | The Jackson Laboratory | RRID:IMSR_JAX:002809 | R6/1 HD model |
| Recombinant DNA reagent | AAV1-CamKIIa-hChR2 (H134H)-eYFP-WPRE.hGH | University of Pennsylvania -Penn Vector Core | Catalog number: AV-1–26969P | titres: $8{,}97 \times 10^{12}$ genomic particles/mL (we used 1:10 dilution) |
| Recombinant DNA reagent | AAV1-CamKIIa(1,3)-eYFP.WPRE.hGH; | University of Pennsylvania -Penn Vector Core | Catalog number: AV-1-PV2975 | titres: $1.18 \times 10^{13}$ genomic particles/mL (we used 1:10 dilution) |
| Software, algorithm | GraphPad Prism software | GraphPad Prism (https://graphpad.com) | RRID:SCR_002798 | Version 8.0.0 |
| Software, algorithm | Fiji | https://imagej.net/Fiji | RRID:SCR_002285 | |
| Software, algorithm | ANTs (Advanced Normalization ToolS) | http://stnava.github.io/ANTs/ | RRID:SCR_004757 | |
| Software, algorithm | FSL | http://www.fmrib.ox.ac.uk/fsl/ | RRID:SCR_002823 | |
| Software, algorithm | Python | http://www.python.org/ | RRID:SCR_008394 | |
| Software, algorithm | SPM | http://www.fil.ion.ucl.ac.uk/spm/ | RRID:SCR_007037 | |
| Software, algorithm | Nitime | http://nipy.org/nitime/ | RRID:SCR_002504 | |
| Software, algorithm | LCModel | http://s-provencher.com/pages/lcmodel.shtml | RRID:SCR_014455 | |
| Software, algorithm | ITK-Snap | http://www.nitrc.org/projects/itk-snap/ | RRID:SCR_002010 | |

## Animals

R6/1 transgenic mice expressing exon-1 of mutant huntingtin containing 115 CAG repeats were acquired from The Jackson Laboratory (Bar Harbor, ME) and maintained in a B6CBA background. Genotypes were obtained by polymerase chain reaction (PCR) from tail biopsy and WT littermates were used as the control group. Animals were housed together in groups of mixed genotypes in a room kept at 19–22°C and 40–60% humidity under a 12:12 hr light/dark cycle with access to water and food ad libitum, and data were recorded for analysis by microchip mouse number. All animal procedures were approved by the animal experimentation Ethics Committee of the Universitat de Barcelona (274/18) and Generalitat de Catalunya (10101), in compliance with the Spanish RD 53/2013 and European 2010/63/UE regulations for the care and use of laboratory animals.

## MRI image acquisition

11 WT mice and 13 R6/1 mice were scanned blind to the experimenter at 17–20 weeks of age on a 7.0T BioSpec 70/30 horizontal animal scanner (Bruker BioSpin, Ettlingen, Germany), equipped with an actively shielded gradient system (400 mT/m, 12 cm inner diameter). Each animal underwent two acquisition sessions. The first session included magnetic resonance spectroscopy (MRS) and CEST (chemical exchange saturation transfer) acquisition to assess metabolite concentration. The second

session was performed one week later and included structural T2-weighted imaging and resting-state functional magnetic resonance (rs-fMRI) to evaluate connectivity between regions of interest. In both cases, animals were placed in a supine position in a Plexiglas holder with a nose cone for administering anesthetic gases and were fixed using tooth and ear bars and adhesive tape. Animals were anesthetized with 2.5% isoflurane (70:30 $N_2O:O_2$). In the second session, a combination of medetomidine (bolus of 0.3 mg/kg, 0.6 mg/kg/h infusion) and isoflurane (0.5%) was used to sedate the animals.

In both sessions, 3D-localizer scans were used to ensure the accurate position of the head at the isocenter of the magnet. During the first session, MRS was acquired using PRESS (voxel size 5.4 μL, TR = 5000 ms, TE = 12 ms, 256 averages, partial water suppression, VAPOR) in a voxel located in the left striatum (*Figure 2a*). Voxel position was defined using T2 RARE images acquired in axial, sagittal, and coronal orientation as reference. A non-suppressed reference water signal was also acquired in the same voxel (eight averages). GluCEST and WASSR (water saturation shift referencing) were acquired from one coronal slice with a slice thickness of 2.5 mm, covering striatum and cortex. WASSR image was acquired as reference using a pulse of 500 ms and 0.5 μT with an offset range from −200 and 200 Hz. GluCEST sequence used a pulse of 6000 ms, 10 μT, and offset ranging from −1600 to 1600 Hz.

In the second session, a T2-weighted image was acquired using a RARE sequence with effective TE = 33 ms TR = 2.3 s, RARE factor = 8, voxel size = 0.08 × 0.08 mm$^2$, and slice thickness = 0.5 mm. rs-fMRI was acquired with an EPI sequence with TR = 2 s, TE = 19.4, voxel size 0.21 × 0.21 mm$^2$, and slice thickness 0.5 mm; 420 volumes were acquired resulting in an acquisition time of 14 min.

## Functional connectivity analysis

Two approaches were used to evaluate functional connectivity. On the one hand, whole-brain connectivity was evaluated using global network metrics (*Rubinov and Sporns, 2010*) and on the other hand, a seed-based analysis was performed to evaluate the connectivity of the striatum with the rest of the brain.

For both approaches, rs-fMRI was preprocessed, including slice timing, motion correction by spatial realignment using SPM8, correction of EPI distortion by elastic registration to the T2-weighted volume using ANTs (*Avants et al., 2008*), detrend, smoothing with a full-width half maximum (FWHM) of 0.6 mm, frequency filtering of the time series between 0.01 and 0.1 Hz and regression by motion parameters. All these steps were performed using NiTime (http://nipy.org/nitime).

The MRI-based atlas of the mouse brain (*Ma, 2008*) was considered for brain parcellation. To obtain more specific regions, the sensorimotor cortex defined in the original atlas was manually divided into somatosensory, orbitofrontal (OFC), motor 1 and 2 (M1, M2), cingulate, medial prefrontal cortex (mPFC), and retrosplenial cortex. The atlas template was elastically registered to each subject T2-weighted volume using ANTs (*Avants et al., 2008*), and the resulting transformation was applied to the label map to obtain the individual brain parcellation.

Region parcellation was registered from the T2-weighted volume to the preprocessed mean rs-fMRI. The whole-brain functional brain network was estimated considering the gray matter regions obtained by parcellation as the network nodes. Connectivity between each pair of regions was estimated as the Fisher-z transform of the correlation between average time series in each region. Network organization was quantified using graph metrics, namely, strength, global, and local efficiency, and average clustering coefficient (*Rubinov and Sporns, 2010*).

To perform the seed-based analysis, the striatum was selected from the automatic parcellation. The average time series in the seed was computed and correlated with each voxel time series, resulting in a correlation map describing the connectivity of striatum with the rest of the brain. The following regions of interest were identified from brain parcellation to evaluate their connectivity with the seed region: striatum, globus pallidus, thalamus, and cingulate, somatosensory cortices, M1, M2, mPFC, and OFC cortices. Connectivity was quantified as the mean value of the correlation map in each region, considering only positive correlations.

## Metabolite quantification

LC-Model (*Provencher, 1993*) was used to quantify the MRS-detectable metabolites in the striatum of mice, fitting a basis set of 19 small metabolites and nine macromolecules. Quality criteria were

defined for the whole spectrum as the FWHM lower than 0.06 ppm and the signal-to-noise ratio greater than 5. Also, metabolites quantified with relative Cramér-Rao lower bounds (CRLB) higher than 15% were discarded.

## GluCEST analysis

To assess glutamate from CEST acquisition, in-home scripts in python were developed to process and quantify GluCEST asymmetry. GluCEST contrast was measured as the asymmetry between the image obtained with saturation at three ppm downfield from water (resonant frequency of exchangeable protons for glutamate) and the image with saturation at −3 ppm as follows (*Cai et al., 2013*; *Cai et al., 2012*):

$$GluCEST_{assym} = \frac{100 \times M_{sat(-3ppm)} - M_{sat(3ppm)}}{M_{sat(-3ppm)}}$$

To quantify and compare GluCEST asymmetry, the ventricles were manually delineated in each GluCEST map and the minimum asymmetry value in this region was taken as a reference value to subtract from the whole-brain map. Average regional GluCEST asymmetry relative to ventricles was computed in manually delineated regions (*Figure 2d*), namely, right and left striatum, and right and left cortex values were averaged.

## Stereotaxic surgery

Stereotaxic surgery was performed in 16-week-old mice under isofluorane anesthesia (5% induction, and 1.5% maintenance). For optogenetic modulation, an adeno-associated virus (AAV) containing Channelrhodopsin (ChR2) under CaMKII promotor was injected (AAV1-CaMKIIa-hChR2(H134H)-eYFP-WPRE.hGH; AAV-ChR2) and a control YFP construct (AAV1-CaMKIIa-eYFP-WPRE.hGH; AAV-YFP) was used as control. Virus production, amplification, and purification were performed by University of Pennsylvania-Penn Vector Core (titers: ~1 × 10$^{12}$ genomic particles/mL) and a volume of 0.5 µL of corresponding viral constructs was injected bilaterally in M2 cortex at the following coordinates: +2.46 anteroposterior (AP), ±1 mediolateral (L), and −0.8 mm dorsoventral (DV) from bregma and dura mater, using 5 µL Hamilton syringe with a 33 gauge needle at 0.1 µL/min. The injection needle was left for an additional 5 min period to allow diffusion of virus particles and avoid reflux.

For in vivo optogenetic stimulation, additional fiber-optic cannulas (MFC_200/240–0.22_3.5_ZF1.25_FLT; Doric Lenses) were placed bilaterally in the striatum at AP +0.14, L ± 2.2, DV −3 mm from bregma and dura mater and secured using dental cement. Optogenetic stimulation and behavioral testing were performed 4 weeks later.

For microdialysis experiments, AAV-YFP or AAV-CHR2 virus constructs were injected in the M2 as described above 4 weeks before the microdialysis procedure. After 4 weeks, one concentric dialysis probe equipped with a Cuprophan membrane (1.5 mm long) was implanted in the dorsal striatum at coordinates (in mm, from bregma and skull): AP −1.1 (with a 20° angle); L 2.2; DV - 4.5 from dura mater, in anesthetized mice (isoflurane, 5% induction and 1.5% maintenance). Additional bilateral fiber-optic cannulas were implanted as described above.

## Optogenetic stimulation

The blue light was delivered from 473 nm diode-pumped solid-state blue laser (Laserglow) at 10 Hz, 5 ms pulse width, and ~5 mW (measured at the end of the patchcord) using a custom-made waveform generator (Arduino). For in vivo experiments, mice were habituated to be connected to the fiber optic patch cord (Doric Lenses) before the experiment. Zirconia sleeves (Doric Lenses) were used to connect the patch cord to the fiber-optic cannulas.

For electrophysiological recordings, the fiber optic patch cord was placed above the slice surface.

## In vivo microdialysis

Extracellular glutamate (Glu) levels were measured by in vivo microdialysis as previously described (*López-Gil et al., 2007*). Microdialysis experiments were performed in freely moving mice 24 hr (day 1) and 48 hr (day 2) after surgery. The artificial cerebrospinal fluid (aCSF) was pumped at 1.65 µL/min and dialysate samples were collected every 6 min in micro vials. Following an initial 3 hr

stabilization period, six baseline samples were collected before 1 min optogenetic stimulation (sample 7, day 1) and 5 min stimulation (sample 7, day 2). Glutamate concentration was analyzed by a blind experimenter using an HPLC consisting in a Waters 717plus autosampler and a Waters 600 quaternary gradient pump, and a Phenomenex Gemini 5 μm 100 × 3 mm column. Dialysate samples were pre-column derivatized with OPA reagent by adding 90 μL distilled water to the 10 μL dialysate sample and followed by the addition of 15 μL of the OPA reagent. After 2.5 min reaction, 80 μL of this mixture was injected into the column. Glutamate was detected with a Waters 2475 Fluorescence detector using excitation and emission wavelength of 360 and 450 nm, respectively. Following sample collection, mice were sacrificed, their brains were removed, sectioned, and stained with neutral red to ensure proper probe placement.

## Multi-electrode arrays

Mouse brain sagittal sections were obtained on a vibratome (Microm HM 650 V, Thermo Scientific, Waltham, MA) at 350 μm thickness in oxygenated (95% $O_2$, 5% $CO_2$) ice-cold artificial cerebrospinal fluid (aCSF) containing (in mM) 124 NaCl, 24 $NaHCO_3$, 13 glucose, 5 HEPES, 2.5 KCl, 2.5 $CaCl_2$, 1.2 $NaH_2PO_4$. 1.3 $MgSO_4$. Slices were then transferred to an oxygenated 32°C recovery solution of the following composition (in mM): 92 NMDG, 30 $NaHCO_3$, 25 glucose, 20 HEPES, 10 $MgSO_4$, 5 sodium ascorbate, 2.5 KCl, 1.2 $NaH_2PO_4$, 3 sodium pyruvate, 2 thiourea, and 0.5 $CaCl_2$; for 15 min (*Choi et al., 2019*). Then, slices were transferred to oxygenated aCSF at room temperature and left for at least 1 hr before recording.

Following recovery, slices were placed in a multi-electrode array (MEA) recording dish, fully submerged in oxygenated aCSF at 37°C. Electrophysiological data were recorded by a blind experimenter with MEA set-up from Multi Channel Systems MCS GmbH (Reutlingen, Germany) composed of 60 channels USB-MEA60-inv system with a blanking unit from Multi Channel Systems and the STG4004 current and voltage generator. Experiments were carried out with 60MEA200/30iR-ITO MEA dishes consisting of 60 planar electrodes arranged in an 8 × 8 array (200 μm distance between neighboring electrodes and an electrode diameter of 30 μm). The software for stimulation, recording, and signal processing were MC Stimulus and MC Rack from Multi Channel Systems. Using a digital camera during recording assessed the position of the brain slices on the electrode field and the location of the laser for stimulation.

Striatal field post-synaptic currents (fPSC) were recorded in the dorsal striatum in response to the stimulation of cortical afferents with short laser pulses of increasing intensity. Light stimulation was performed by placing the laser in the slice approximately 0.5 mm posterior to bregma, 2.5 mm ventral from the surface, and 3 mm lateral from the midline (*Figures 3d–e* and *6a*). The evoked striatal fPSCs were analyzed after trains of 1 ms light stimuli of increasing intensity (0.00, 0.08, 1.5, 5.6, 13.7, 28.7, 47.7, and 60 mW/mm$^2$).

For electrical stimulation, we set one MEA electrode located approximately at 1 mm anterior to bregma, 2.5 mm depth from the brain surface and 2.2 mm lateral from the midline as stimulation one (*Figure 6a*, panel right), and fPSC were evoked by single monopolar biphasic pulses (negative/positive, 100 μs per phase). For input/output curves, cortico-striatal fibers were stimulated with trains of three positive-negative identical pulses at increasing currents (5 s inter-pulse time; 30 s train intervals; 250–3000 μV currents). Then, the pulse amplitude of subsequent stimuli was set to evoke 40% of the saturating fPSC response in the input/output curve. Long-term depression (LTD) was induced by theta-burst stimulation (TBS) consisting of 10 trains spaced 15 s apart. Each train consisted of 10 bursts at 10.5 Hz (theta), and each burst consisted of four stimuli at 50 Hz. Thus, the whole TBS stimulation period lasted 2.5 min (*Hawes et al., 2013*). In all cases, the amplitude of the evoked fPSC was quantified. Software for recording and signal processing was MC Rack from Multi Channel Systems. Using a digital camera during recording assessed the position of the brain slices on the electrode field.

## Behavioral assessment

All behavioral tests were performed as previously described (*Creus-Muncunill et al., 2019*; *Puigdellívol et al., 2015*). Tests were done during the light phase and animals were habituated to the experimental room for at least 1 hr before testing. The apparatus was thoroughly cleaned between tests and animals. The order of the tests and time between tests is detailed in *Figure 4d*.

## Open field

Spontaneous locomotor activity and exploratory behavior were assessed in an open field (OF) at weeks 20, 21, and 22 of age. The OF test was performed using a white square arena (40 × 40 × 30 cm$^3$) with dim light (~20 lux). Mice were left in the center of the apparatus and allowed to explore the arena for 11 min. Optogenetic stimulation was performed from 5 to 6 min of the test. Animals were tracked and recorded with SMART 3.0 software (Panlab). The number of events and time spent doing rearing and grooming were measured to assess exploratory and stereotypic behavior.

## Accelerating rotarod

Mice were placed on a motorized rod (30 mm diameter) with a rotation speed gradually increased from 4 to 40 rpm over 5 min and the latency to fall was recorded to assess motor learning (*Creus-Muncunill et al., 2019*). The procedure was performed four times per day with 1 hr inter-trial interval for 3 consecutive days, 12 trials in total.

## Balance beam

Motor coordination and balance were determined by measuring the ability of the mice to traverse a narrow beam according to *Creus-Muncunill et al., 2019*. The beam consisted of a wooden square bar (50 cm long with 1.3 cm face), divided by 5 cm-frames and placed horizontally 50 cm above the bench surface, with each end mounted on narrow support. Animals could walk for 2 min along the beam and the number of frames crossed was measured.

## Golgi stain and dendritic spine quantification

Fresh brain hemispheres were processed using the Rapid GolgiStain Kit (FD Neurotechnologies), as previously described (*Alvarez-Periel et al., 2018*). Bright-field images Golgi-impregnated MSNs striatal neurons from 100 μm brain slices were captured by a blind experimenter using a Leica epi-fluorescence microscope (×63 oil objective, magnification 1,6). Image Z stacks were taken every 0.2 μm and analyzed with Fiji software. Dendritic segments were traced (>20 μm long; average, 47.35 μm; mean range, 20–95 μm), and the spines were counted. Spine density was calculated in three to thirteen dendrites per neuron, and the values were averaged to obtain the mean for each neuron (7–11 different neurons per animal, n = 3 for WT-YFP, WT-ChR2, and HD-YFP, and n = 2 for HD-ChR2).

## Statistical analysis

All data are expressed as mean ± SEM. Each point represents data from an individual mouse. Statistical analysis was performed using the two-tailed unpaired Student's t-test or two-way ANOVA, and Bonferroni post hoc test as indicated in the figure legends. Values of $p < 0.05$ were considered as statistically significant.

# Acknowledgements

We are very grateful to Ana López (María de Maeztu Unit of Excellence, Institute of Neurosciences, University of Barcelona, MDM-2017–0729, Ministry of Science, Innovation, and Universities) and Maite Muñoz for their excellent technical support. This work has been funded by the Spanish Ministry of Sciences, Innovation and Universities through projects no. SAF2017-88076-R, RETICS de Terapia Celular, Instituto de Salud Carlos III (RD06/0010/0006), and la Marató de TV3. This research is part of NEUROPA. The NEUROPA Project has received funding from the European Union's Horizon 2020 Research and Innovation Program under Grant Agreement No. 863214.

# Additional information

## Funding

| Funder | Grant reference number | Author |
| --- | --- | --- |
| Spanish Ministry of Sciences, Innovation and Universities | SAF2017-88076-R | Jordi Alberch |
| Instituto de Salud Carlos III | RETICS (RD06/0010/0006) | Jordi Alberch |

| Horizon 2020 - Research and Innovation Framework Programme | Grant Agreement No. 863214 | Mercè Masana |
|---|---|---|
| Fundació la Marató de TV3 | | Jordi Alberch |
| María de Maeztu Unit of Excellence, Institute of Neurosciences, University of Barcelona, Ministry of Science, Innovation and Universities | MDM-2017-0729 | Jordi Alberch |

The funders had no role in study design, data collection and interpretation, or the decision to submit the work for publication.

### Author contributions

Sara Fernández-García, Data curation, Formal analysis, Funding acquisition, Validation, Investigation, Methodology, Writing - original draft, Writing - review and editing; Sara Conde-Berriozabal, Data curation, Formal analysis, Funding acquisition, Investigation, Methodology, Writing - review and editing; Esther García-García, Data curation, Funding acquisition, Methodology, Writing - review and editing; Clara Gort-Paniello, Investigation, Methodology, Writing - review and editing; David Bernal-Casas, Data curation, Supervision, Validation, Investigation, Writing - review and editing; Gerardo García-Díaz Barriga, Conceptualization, Investigation, Methodology; Javier López-Gil, Investigation, Methodology; Emma Muñoz-Moreno, Data curation, Software, Formal analysis, Validation, Writing - review and editing; Guadalupe Soria, Software, Supervision, Validation, Writing - review and editing; Leticia Campa, Formal analysis, Methodology, Writing - review and editing; Francesc Artigas, Resources, Supervision, Funding acquisition, Writing - review and editing; Manuel José Rodríguez, Data curation, Formal analysis, Validation, Investigation, Methodology, Writing - review and editing; Jordi Alberch, Conceptualization, Resources, Supervision, Funding acquisition, Validation, Project administration, Writing - review and editing; Mercè Masana, Conceptualization, Resources, Data curation, Formal analysis, Supervision, Funding acquisition, Validation, Investigation, Methodology, Writing - original draft, Writing - review and editing

### Author ORCIDs

Sara Fernández-García https://orcid.org/0000-0003-2281-4526
Sara Conde-Berriozabal https://orcid.org/0000-0002-3712-8046
Esther García-García https://orcid.org/0000-0003-3997-493X
Clara Gort-Paniello https://orcid.org/0000-0003-3850-918X
Emma Muñoz-Moreno https://orcid.org/0000-0003-2104-7265
Guadalupe Soria https://orcid.org/0000-0002-8344-0479
Manuel José Rodríguez https://orcid.org/0000-0002-4476-004X
Jordi Alberch https://orcid.org/0000-0002-8684-2721
Mercè Masana https://orcid.org/0000-0003-1392-4774

### Ethics

Animal experimentation: All animal procedures were approved by the animal experimentation Ethics Committee of the Universitat de Barcelona (274/18) and Generalitat de Catalunya (10101), in compliance with the Spanish RD 53/2013 and European 2010/63/UE regulations for the care and use of laboratory animals.

### Decision letter and Author response

Decision letter https://doi.org/10.7554/eLife.57017.sa1
Author response https://doi.org/10.7554/eLife.57017.sa2

## Additional files

### Supplementary files

• Supplementary file 1. Average seed-based BOLD correlation maps from WT mouse striatum related to *Figure 1b*, visualized with ITK-SNAP software (*Yushkevich et al., 2006*).

• Supplementary file 2. Average seed-based BOLD correlation maps from R6/1 HD mouse striatum related to *Figure 1b*, visualized with ITK-SNAP software (*Yushkevich et al., 2006*).

• Transparent reporting form

### Data availability

All data generated or analysed during this study are included in the manuscript and supporting files.

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
