## [Decision Letter]

**Acceptance summary:**

This study uses a range of techniques to demonstrate reduced brain connectivity between the cortex and a subcortical region in a model of Huntington's Disease. The study shows that selective stimulation of this connection improved the disrupted motor behavior and neurobiological adaptations associated with Huntington's Disease. These findings implicate a potential brain circuit based therapy for Huntington's Disease.

**Decision letter after peer review:**

Thank you for submitting your article "M2 Cortex-Dorsolateral striatum stimulation reverses motor symptoms and synaptic deficits in Huntington's Disease" for consideration by *eLife*. Your article has been reviewed Kate Wassum as the Senior Editor, a Reviewing Editor, and three reviewers. The reviewers have opted to remain anonymous.

The reviewers have discussed the reviews with one another and the Reviewing Editor has drafted this decision to help you prepare a revised submission.

Summary:

The authors combine a wide array of techniques to investigate a circuit-based therapy for Huntington's Disease (HD). Specifically, they focus on a deficit in frontostriatal connectivity, M2 cortex-dorsolateral striatum, and glutamate release and how this deficit might be rescued by optogenetic stimulation. The study reports decreased overall connectivity between cortex and striatum in the R6/1 mouse model of HD during the symptomatic stage. Specifically, the study implicates changes in structural and synaptic plasticity in the dorsolateral striatum, and that optogenetic stimulation of this circuit reverses those changes. Overall, the study provides new insight into cortical striatal disruption in HD.

Essential revisions:

1) While no new experiments are required it is necessary to reframe the conclusions to support the results and/or provide additional statistical analyses to support some of the conclusions. See specific comments below. Additionally, please adjust the nomenclature when stating "frontal cortex" and contrasting with "motor cortex" and "cingulate cortex." Use of frontal cortex (as in Figure 1) appears to refer only to M2, but this is unclear and perhaps the authors are also including OFC, PL, IL, etc). Another concern is that the motor cortex is part of the frontal cortex, thus more specific nomenclature should be used.

2) Please clarify the analyses presented in subsection “Frontal cortex, but not motor cortex, show diminished functional connectivity with the striatum during rest in HD mice”. For example, for the second two-way ANOVA, the authors refer to "correlation values between left motor cortex and the same basal ganglia nuclei." But then the follow-up refers to frontal cortex, sensorimotor cortices cingulate cortices, and thalamus-not basal ganglia nuclei. Also, although correct, it would be easier to understand if "correlation values" were replaced with "functional connectivity" (since there is a two-way ANOVA to track as well). Finally, the summary focuses on cortico-striatal pathways only, whereas the results up until that point were heavily weighted to cortico-cortical and even cortical-thalamic connectivity.

3) The authors claim that an imbalance arises in functional BOLD connectivity between frontostriatal and motor-corticostriatal circuits and they state that "different cortico-striatal pathways are distinctly altered in HD mice,". Although the reduction in motor cortex (M1) in R6/1 mice does not reach significance in Figure 1, it does not appear different than the reduction in frontostriatal connectivity. Further, the correlation between frontal cortex (M2) and striatum just reached significance (p=0.0447). Please provide additional post-hoc analysis to show statistically that the differences (between R6/1 and WT) are significantly different (between frontal-striatal and motor-striatal). This has implications for the M2 vs M1 distinction which should be clarified in the Discussion section. In the current form the results do not support the conclusions.

4) As the authors mention in the Introduction, the striatum receives glutamatergic inputs from all cortical areas, thus it is likely that other cortical regions outside of the M2 region are involved in the motor deficits. Please provide caution when suggesting that selective stimulation of the frontal cortex (M2)-striatal pathway is what improves, almost exclusively, behavioral, functional and morphological outcomes. In the current form the M2 afferent specificity is presented as a conceptual advance. To support this conclusion the authors would need to demonstrate their optogenetic manipulations are specific for the frontostriatal, rather than motor-corticostriatal circuit, especially since their primary behavioral readout is motor function. Since the selective stimulation of the primary motor cortex (M1)-striatal pathway is not included then the conclusions focused on M2 should be reframed. Additionally, please provide rationale as to why the same optical stimulation protocol to test the effects of the M1-striatal pathway on the HD phenotype, at least as a control to demonstrate differential effects?

5) Please clarify if the entire cerebral cortex or just the frontal and motor cortices were used in the MRS analysis.

6) The conclusion of decreased striatal glutamate occurring in R6/1 mice is based on a small difference in the CEST measurements. Further, this finding was not replicated using two other independent techniques (MRS and microdialysis [at baseline]). Thus, in the current form the claim of altered striatal glutamate in R6/1 mice is not supported. Please adjust the conclusions to reflect the present data.

7) Using microdialysis the authors show that optogenetic stimulation of M2 corticostriatal afferents leads to no detectable glutamate release in R6/1 mice. If optogenetically stimulating of M2 afferents in R6/1 mice does not lead to glutamate release in the striatum, how do the authors propose their optogenetic manipulations are improving striatal function? Clearly some glutamate must be released in order to detect fEPSCs in Figure 3.

8) Please also clarify why glutamate release and field potential measurements where not performed with M1 stimulation?

9) In Figure 3A, since the injection was in M2, please add a little syringe going into that region. This will help the reader.

10) The result in Figure 3C is quite striking. To determine that the M2-DLS pathway was targeted in both WT and R6/1 please provide histological images.

11) The spread of the injection site and resulting terminal fields and fibers in Figure 4C appears far outside of area M2 into other frontal regions. Please comment on this and the implications this has on the data interpretations.

12) The authors claim that the effects of optogenetic manipulation on open-field behavior are long-lasting (up to one week), but the analysis to support this conclusion was not performed. Instead, only summary statistics on the full 11-minutes of open-field exploration is reported. Please report comparisons of the effects from the 5-minute period of pre-stimulation across days to exclude the effects of the stimulation and immediate post-stimulation periods. In the post-hoc analysis (S4), none of those time-points are labeled significant.

13) Please clarify the focus on LTD, when rotarod performance is associated with LTP in the DLS (they do cite Yin, 2009). Moreover, this type of LTD appears to be largely intact in the R6/1 mice. The fact that LTD is modestly improved by optogenetic stimulation here is ambiguous, given there was no significant LTD deficit to begin with.

14) It is interesting that fiber tracts were not differentially affected, even though the spine density of MSNs showed significant spine loss. If, as the authors contend, the fronto-striatal pathway is more affected than the motor-striatal pathway, one would have expected differences in fiber tracts. Please discuss these conflicting findings.

15) Please clarify and rephrase the following sentence in the Discussion section as it is not clear in the present form. "Our results indicate that cortical afferences rather than output pathways are predominantly affected during disease progression, highlighting that frontal cortex, rather than primary motor cortex, has a stronger contribution to the basal ganglia network dysfunction in HD".

[Editors' note: further revisions were suggested prior to acceptance, as described below.]

Thank you for resubmitting your article "M2 Cortex-Dorsolateral striatum stimulation reverses motor symptoms and synaptic deficits in Huntington's Disease" for consideration by *eLife*. Your revised article has been reviewed by Kate Wassum as the Senior Editor, a Reviewing Editor, and three reviewers. The following individual involved in review of your submission has agreed to reveal their identity: Carlos Cepeda (Reviewer #1).

The reviewers have discussed the reviews with one another and the Reviewing Editor has drafted this decision to help you prepare a revised submission.

Summary:

In this study, Fernandez-Garcia et al., attempted to dissect specific cortico-striatal pathways affected in Huntington's disease (HD). In particular, they examined fronto-striatal and primary motor-striatal connectivity in the R6/1 mouse model during the symptomatic stage (around 17-20 weeks of age). Using a wide array of methodologies, including in vivo functional MRI, Magnetic Resonance Spectroscopy (MRS), optogenetics coupled with in vivo microdialysis, ex vivo multielectrode arrays, GluCEST, Golgi staining, and a number of behavioral tests, they report decreased overall connectivity between cortex and striatum, but the effect was particularly evident in the M2-dorsolateral striatal projection. They also found reduced glutamate release onto the striatum. Notably, repeated optogenetic stimulation of M2 rescued corticostriatal connectivity and ameliorated the HD phenotype including behavior, synaptic plasticity, and spine density. The authors conclude that the selective stimulation of fronto-striatal pathways can become an effective therapeutic strategy in HD.

While progress was made in the revision there still remain concerns with respect to points 1-4 of the previous reviews and data in Figure 6. Thus, additional revisions are necessary.

Essential revisions:

1) The concern with the nomenclature addressed in comment #1 still requires improvement. M2 is split between "frontal" areas and "motor" areas, and these results form the backbone of the claim that it is M2-striatal projections that are affected. Please clarify why the authors did not perform an analysis on M2-striatal vs M1-striatal vs CG-striatal. Additionally, please clarify why they are grouped in this unusual manner when M2 is the focus of the paper. Some further analyses may also help address this concern.

2) With respect to comment #2 the terminology requires updating in the figures as it was only updated in the text.

3) With respect to comment #3 the authors have not adequately addressed this criticism. It is insufficient to state that frontal-striatal correlation is higher in WT than HD, but that this does not achieve significance in motor cortex. If the p-value of one reached significance in post-hoc analysis but another did not this does not imply that the two are statistically different (https://www.nature.com/articles/nn.2886). Please show that the difference between WT and HD is significantly greater for frontal than motor cortex. There are a number of ways to achieve this, including showing a significant interaction effect, performing a fisher r to z on the p values, calculating difference scores for each individual and running a paired-samples t-test, etc. Alternatively, the authors could remove Figure 1, de-emphasize this specificity, acknowledge that several brain regions have disrupted functional connectivity and that they looked closer at one of these brain regions.

4) With respect to comment #4 the Discussion section and Abstract are still problematic on these points. The study lacks evidence that "frontal areas show stronger deficits in functional connectivity with basal ganglia related nuclei in our HD mice," because that statistical test was not performed. Please amend this further.

5) One additional point is that the authors should determine why in Figure 6B, there was no reduction in light-evoked EPSPs in HD mice, when this the exact same experiment as in Figure 3E, where the authors claim there is a significant reduction in HD mice? Figure 3E was the strongest evidence the authors had for a deficit in PFC to DLS glutamate transmission, and it is refuted by Figure 6B. Overall, along with the lack of a significant deficit in LTD (Figure 6d) there are concerns with the scientific rigor. Along these same lines, please provide information about experimenter blinding, which could have biased analyses, particularly of small effect sizes on behavior or spine quantification.

[Editors' note: further revisions were suggested prior to acceptance, as described below.]

Thank you for resubmitting your work entitled "M2 Cortex-Dorsolateral striatum stimulation reverses motor symptoms and synaptic deficits in Huntington's Disease" for further consideration by *eLife*. Your revised article has been evaluated by Kate Wassum (Senior Editor) and a Reviewing Editor.

The manuscript has been improved but there are some remaining issues that need to be addressed before acceptance, as outlined below:

Please update Figure 1C to include a y-axis and clarify that this figure reflects the strength of the correlation with predefined seed regions and not the number or proportion of voxels within a seed region with significant correlation (as in Figure 1B).

---

## [Author Response]

Essential revisions:1) While no new experiments are required it is necessary to reframe the conclusions to support the results and/or provide additional statistical analyses to support some of the conclusions. See specific comments below. Additionally, please adjust the nomenclature when stating "frontal cortex" and contrasting with "motor cortex" and "cingulate cortex." Use of frontal cortex (as in Figure 1) appears to refer only to M2, but this is unclear and perhaps the authors are also including OFC, PL, IL, etc). Another concern is that the motor cortex is part of the frontal cortex, thus more specific nomenclature should be used.

We agree with the reviewer that the nomenclature used could lead to misinterpretations. To avoid any confusion, we added the regions of interest obtained by atlas-based parcellation that we used for the analysis in Figure 1. In Figure 1 and Figure 1—figure supplement 1, frontal cortex includes OFC, prelimbic, infralimbic, M2 and frontal association areas, while motor cortex include more caudal M1/M2 regions, and this information is now added in the Results section.

2) Please clarify the analyses presented in subsection “Frontal cortex, but not motor cortex, show diminished functional connectivity with the striatum during rest in HD mice”. For example, for the second two-way ANOVA, the authors refer to "correlation values between left motor cortex and the same basal ganglia nuclei." But then the follow-up refers to frontal cortex, sensorimotor cortices cingulate cortices, and thalamus-not basal ganglia nuclei. Also, although correct, it would be easier to understand if "correlation values" were replaced with "functional connectivity" (since there is a two-way ANOVA to track as well). Finally, the summary focuses on cortico-striatal pathways only, whereas the results up until that point were heavily weighted to cortico-cortical and even cortical-thalamic connectivity.

The reviewer is right pointing that thalamus and cortex are not basal ganglia nuclei and we corrected the text in the Results section accordingly. We also changed correlation values to functional connectivity to allow better understanding as suggested.

Finally, we agree that our data indicate strong cortico-cortical and cortical-thalamic connectivity deficits from both frontal and motor cortices, and we included a new sentence in the Results section and Discussion section with this information. We also added that post hoc analysis failed to show significant differences in functional connectivity between the motor cortex and the striatum in order to highlight the differential alteration of fronto-striatal and motor-striatal functional connectivity in HD.

3) The authors claim that an imbalance arises in functional BOLD connectivity between frontostriatal and motor-corticostriatal circuits and they state that "different cortico-striatal pathways are distinctly altered in HD mice,". Although the reduction in motor cortex (M1) in R6/1 mice does not reach significance in Figure 1, it does not appear different than the reduction in frontostriatal connectivity. Further, the correlation between frontal cortex (M2) and striatum just reached significance (p=0.0447). Please provide additional post-hoc analysis to show statistically that the differences (between R6/1 and WT) are significantly different (between frontal-striatal and motor-striatal). This has implications for the M2 vs M1 distinction which should be clarified in the Discussion section. In the current form the results do not support the conclusions.

We agree with the reviewer that in Figure 1 the reduction levels of functional connectivity between frontal or motor cortex and striatum look quite similar and indeed we had previously evaluated different tests to ensure the reported results. Note that with the answer in point 1, we clarified that frontal cortex does not only include M2, and motor cortex includes M1 and M2 cortices.

We performed 2-way ANOVA and used Sidak post hoc test, which is another strict test counteracting the family wise error rate in multiple comparisons, and according to Graphpad Statistics Guide, has a bit more power. Using the Sidak post hoc test we found similar results as the ones reported: Frontal cortex showed reduced functional connectivity with left motor cortex (p = 0.0186), somatosensory cortex (p = 0.0071), cingulate cortex (p = 0.0003), striatum (p = 0.0438), and with the thalamus (p= 0.0077); while motor cortex showed reduced connectivity with left frontal (p= 0.0175), somatosensory (p= 0.0186) and cingulate cortices (p= 0.0041) and the thalamus (p= 0.0311) but not striatum (p=0.0803) nor globus pallidus (p= 0.75).

Then, for consistency with the rest of the manuscript, we prefer to leave the Bonferroni post-hoc test.

4) As the authors mention in the Introduction, the striatum receives glutamatergic inputs from all cortical areas, thus it is likely that other cortical regions outside of the M2 region are involved in the motor deficits. Please provide caution when suggesting that selective stimulation of the frontal cortex (M2)-striatal pathway is what improves, almost exclusively, behavioral, functional and morphological outcomes. In the current form the M2 afferent specificity is presented as a conceptual advance. To support this conclusion the authors would need to demonstrate their optogenetic manipulations are specific for the frontostriatal, rather than motor-corticostriatal circuit, especially since their primary behavioral readout is motor function. Since the selective stimulation of the primary motor cortex (M1)-striatal pathway is not included then the conclusions focused on M2 should be reframed. Additionally, please provide rationale as to why the same optical stimulation protocol to test the effects of the M1-striatal pathway on the HD phenotype, at least as a control to demonstrate differential effects?

We agree with the reviewers that other cortical regions outside of M2 might be involved in the motor deficits and with the present results we cannot exclude that stimulation of other cortico-striatal circuits could also improve symptoms in HD. Thus, we rephrased the Results section and Discussion section to highlight frontal and particularly M2 cortex-DLS deficits in HD, while not underestimating the role of M1 or other cortical subregions.

Although the present work did not attempt to decipher the different contribution of M1 and M2 to the HD pathology, in order to compare the effects of M1 or M2 afferent stimulation in the DLS, injection of AAV should be located in M1, while fiberoptic cannula should be placed in same place in the DLS.

5) Please clarify if the entire cerebral cortex or just the frontal and motor cortices were used in the MRS analysis.

While MRS study was only performed in the striatum, in the case of GluCEST, measures were performed in the illustrated 2 mm section containing striatum and cortex. We added a new representative section with the regions of interest labelled for clarification this information is added in Figure 2 for clarification.

6) The conclusion of decreased striatal glutamate occurring in R6/1 mice is based on a small difference in the CEST measurements. Further, this finding was not replicated using two other independent techniques (MRS and microdialysis [at baseline]). Thus, in the current form the claim of altered striatal glutamate in R6/1 mice is not supported. Please adjust the conclusions to reflect the present data.

The reviewer is right pointing this out. While we wanted to stress that glutamate transmission is consistently impaired, there is still controversy regarding the total absolute values of glutamate in the striatum, which includes non-synaptic sources (Ramadan et al., 2013; van der Zeyden et al., 2008). Therefore, we corrected the title of the corresponding Results section, and adjusted the discussion with the presented results, as suggested.

7) Using microdialysis the authors show that optogenetic stimulation of M2 corticostriatal afferents leads to no detectable glutamate release in R6/1 mice. If optogenetically stimulating of M2 afferents in R6/1 mice does not lead to glutamate release in the striatum, how do the authors propose their optogenetic manipulations are improving striatal function? Clearly some glutamate must be released in order to detect fEPSCs in Figure 3.

We agree that some glutamate must be released from HD mice upon optogenetic stimulation, but because glutamate baseline levels measured by microdalysis include glutamate from non-synaptic sources, small increases in glutamate release might not be detected. We changed the sentence in results and added this information in the Discussion section accordingly.

8) Please also clarify why glutamate release and field potential measurements where not performed with M1 stimulation?

From the fronto-striatal pathway shown to be altered in HD, we chose to stimulate the projection from rostral M2 Cortex to DLS based on the structural alterations previously reported (Hintiryan et al., 2016). This information is added in the Results section.

9) In Figure 3A, since the injection was in M2, please add a little syringe going into that region. This will help the reader.

According to the reviewer’s suggestion, we added the syringe in Figure 3A and also Figure 3D, Figure 4A and Figure 5A.

10) The result in Figure 3C is quite striking. To determine that the M2-DLS pathway was targeted in both WT and R6/1 please provide histological images.

We added new histological images in Figure 3 where AAV injection is observed in M2, location of optogenetic fiberoptic canula is in the DLS, and microdialysis probe is just below the fiberoptic canula, for both WT and HD mice.

11) The spread of the injection site and resulting terminal fields and fibers in Figure 4C appears far outside of area M2 into other frontal regions. Please comment on this and the implications this has on the data interpretations.

We understand that YFP expression in Figure 4C could lead to misinterpretation, as both somas and terminal fibers express YFP. For this reason we added in the result section further information to clarify that YFP somas of neurons expressing YFP where located in M2 Cortex injection site and, the presence of fluorescent fiber bundles and axons where found in cortical, striatal and other subcortical projecting areas previously described (Hintiryan et al., 2016; Reiner et al., 2010; Shepherd, 2013) region (Figure 4C; Figure 4—figure supplement 1). Fiberoptic cannulas were placed caudal in the dorsolateral striatum, avoiding direct activation of passing fibers expressing the AAV constructs.

12) The authors claim that the effects of optogenetic manipulation on open-field behavior are long-lasting (up to one week), but the analysis to support this conclusion was not performed. Instead, only summary statistics on the full 11-minutes of open-field exploration is reported. Please report comparisons of the effects from the 5-minute period of pre-stimulation across days to exclude the effects of the stimulation and immediate post-stimulation periods. In the post-hoc analysis (S4), none of those time-points are labeled significant.

We added new Figure 4—figure supplement 3 evaluating the effects between groups during the 5 minutes previous and the 5 minutes posterior to the optogenetic stimulation, as suggested. Please also note that in Figure 4—figure supplement 2 only post hoc analysis is shown when there was group and time interaction 2-way ANOVA effects. Because Bonferroni post hoc take into account multiple comparisons, some of the effects might not reach significance here.

13) Please clarify the focus on LTD, when rotarod performance is associated with LTP in the DLS (they do cite Yin, 2009). Moreover, this type of LTD appears to be largely intact in the R6/1 mice. The fact that LTD is modestly improved by optogenetic stimulation here is ambiguous, given there was no significant LTD deficit to begin with.

Although rotarod performance might also involve LTP as indicated, in the context of Huntington Disease several publications including the ones from our group, showed an impairment in LTD in R6/1 mice. We added new references not from our group to strengthen this point (Cummings et al., 2006; Ghiglieri et al., 2019; Li et al., 2015). Regarding our results here, LTD was induced in R6/1 mice, but to a lesser extent than WT mice. If we performed a Student t-test between WT-YFP and HD-YFP we found statistical significance (p=0.0150). However, because the number of mice and groups evaluated here, this specific comparison did not reach statistical significance after 2-way ANOVA and Bonferroni post hoc comparisons. We added all this information in the Results section for clarity.

14) It is interesting that fiber tracts were not differentially affected, even though the spine density of MSNs showed significant spine loss. If, as the authors contend, the fronto-striatal pathway is more affected than the motor-striatal pathway, one would have expected differences in fiber tracts. Please discuss these conflicting findings.

Fiber tracts obtained from MRI studies relate to thick bundles of axons in the brain (we added a sentence for clarification). Bundles of axons are usually packed together and the water inside diffuse preferentially in one direction (longitudinally) but not transversally to the axon path. While the increase in spine density could be related to an increase in water diffusion in other directions, and therefore a decrease in anisotropy, it is important to note that the MRI resolution is 0.21x0.21x0.5mm³, while the dendritic segments that were traced are in average 37 microns long, and therefore MRI could not be sensitive enough to be influenced by the observed differences in spine density. In addition, since average FA is computed in the whole tract, even in the case that MRI would be sensitive to local spine density loss, this effect might not significantly affect the average FA value.

15) Please clarify and rephrase the following sentence in the Discussion section as it is not clear in the present form. "Our results indicate that cortical afferences rather than output pathways are predominantly affected during disease progression, highlighting that frontal cortex, rather than primary motor cortex, has a stronger contribution to the basal ganglia network dysfunction in HD".

We thank the reviewer for the observation, and re-phrased the sentence to:

“Our results indicate general cortico-cortical and cortico-thalamic functional connectivity deficits in HD mice, suggesting that cortical afferents rather than striatal output pathways are predominantly affected during disease progression. In addition, our data highlight that cortico-striatal pathways are distinctly altered in HD depending on the cortical subregion where they originate. Particularly, frontal areas strongly contribute to the basal ganglia network dysfunction in our HD mice.”

[Editors' note: further revisions were suggested prior to acceptance, as described below.]

While progress was made in the revision there still remain concerns with respect to points 1-4 of the previous reviews and data in Figure 6. Thus, additional revisions are necessary.

We thank the reviewer for all the additional comment which indeed help to further improve the present manuscript.

Essential revisions:1) The concern with the nomenclature addressed in comment #1 still requires improvement. M2 is split between "frontal" areas and "motor" areas, and these results form the backbone of the claim that it is M2-striatal projections that are affected. Please clarify why the authors did not perform an analysis on M2-striatal vs M1-striatal vs CG-striatal. Additionally, please clarify why they are grouped in this unusual manner when M2 is the focus of the paper. Some further analyses may also help address this concern.

As suggested by the reviewer, we performed further analysis to address this concern. We used a new region parcellation of the cortex according to Paxinos atlas, which allowed us to unify nomenclature throughout the manuscript. New labelled regions can be visualized in new Figure 1. We segmented M1, M2, mPFC, OFC, cingulate and somatosensorial cortices, as well as the Globus pallidus and thalamus, according to Paxinos brain regions.

In the new analysis, we placed the seed in the striatum and evaluated the differential functional connectivity between genotypes and between the different brain regions analyzed: M1, M2, mPFC, OFC, cingulate and somatosensorial cortices, thalamus and Globus pallidus. In these new generated data, two-way ANOVA showed genotype and region effects but not genotype/region interaction effect. Bonferroni post-hoc test showed significant functional differences between WT and HD mice in all cortical regions and thalamus but not for Globus pallidus.

We updated Supplementary file 1 and Supplementary file 2 with the new average seed-based BOLD correlation maps from striatum of WT and R6/1 mouse.

We also re-write the title and text of the corresponding Results section according to the new generated data (subsection “The striatum of HD mice shows decreased functional connectivity with afferent regions during rest.”). We also removed previous Figure 1—figure supplement 1 and corresponding Materials and methods section as they do not apply to data generated with the new labels.

2) With respect to comment #2 the terminology requires updating in the figures as it was only updated in the text.

We thank the reviewer for the observation, and we updated the new Figure 1 accordingly.

3) With respect to comment #3 the authors have not adequately addressed this criticism. It is insufficient to state that frontal-striatal correlation is higher in WT than HD, but that this does not achieve significance in motor cortex. If the p-value of one reached significance in post-hoc analysis but another did not this does not imply that the two are statistically different (https://www.nature.com/articles/nn.2886). Please show that the difference between WT and HD is significantly greater for frontal than motor cortex. There are a number of ways to achieve this, including showing a significant interaction effect, performing a fisher r to z on the p values, calculating difference scores for each individual and running a paired-samples t-test, etc. Alternatively, the authors could remove Figure 1, de-emphasize this specificity, acknowledge that several brain regions have disrupted functional connectivity and that they looked closer at one of these brain regions.

We thank the reviewer for the clear explanation regarding the statistical analysis and the suggestions of how to interpret and perform it correctly.

As mentioned in question 1, we performed new analysis with the newly segmented brain regions. Moreover, we placed the seed in the striatum to assess functional connectivity with different brain regions. The statistical analysis of the new generated data showed genotype and region but not interaction effect, as described by two-way ANOVA. Bonferroni post-hoc test showed significant functional differences between WT and HD mice in all regions tested except for Globus pallidus. These new data allows us to conclude that all cortical regions explored showed deficient functional connectivity with the striatum in HD compared to WT (subsection “The striatum of HD mice shows decreased functional connectivity with afferent regions during rest”), and we avoid comparisons/statements about degree of alteration between the different cortico-striatal subcircuits taking into account the reviewer’s comments.

In detail:

-We re-wrote the title and Results section according to the new data provided.

-We removed the emphasis throughout the Discussion section regarding specific cortico-striatal circuits alterations in HD. Specially, we removed the sub-section “fronto striatal circuits show broader functional alterations than motor striatal circuits in HD”, and instead we added a small paragraph indicating that our results indicate general cortico-striatal and thalamo-striatal deficits in functional connectivity under the new subsection “Cortico-striatal transmission is reduced in HD mice”.

-We modified the discussion about specificity of M2-DLS according with the new data in the first and last paragraph of the Discussion section and also Abstract.

4) With respect to comment #4 the Discussion section and Abstract are still problematic on these points. The study lacks evidence that "frontal areas show stronger deficits in functional connectivity with basal ganglia related nuclei in our HD mice," because that statistical test was not performed. Please amend this further.

We amended this concern as indicated in previous answers and considering the new generated data.

5) One additional point is that the authors should determine why in Figure 6B, there was no reduction in light-evoked EPSPs in HD mice, when this the exact same experiment as in Figure 3E, where the authors claim there is a significant reduction in HD mice? Figure 3E was the strongest evidence the authors had for a deficit in PFC to DLS glutamate transmission, and it is refuted by Figure 6B. Overall, along with the lack of a significant deficit in LTD (Figure 6d) there are concerns with the scientific rigor. Along these same lines, please provide information about experimenter blinding, which could have biased analyses, particularly of small effect sizes on behavior or spine quantification.

We realize it is not clear enough that experiments showed in Figure 3E (light-evoked EPSPs in non-previously stimulated mice) are not equivalent to Figure 6B (light-evoked EPSPs in mice that previously received repetitive corticostriatal stimulation). Thus, we added a scheme in Figure 6 and a sentence (subsection “Repeated cortico-striatal stimulation triggers persistent improvements of synaptic plasticity in symptomatic HD mice”) for clarity. Also, MEA recordings were performed blindly and by a different researcher than the one who performed the optogenetic stimulation and behavioral characterization. Information regarding the blindness of the analysis and experimental performance is now added in the manuscript’s Materials and methods section.

[Editors' note: further revisions were suggested prior to acceptance, as described below.]

The manuscript has been improved but there are some remaining issues that need to be addressed before acceptance, as outlined below:

We are glad that reviewers found that the manuscript has been improved, and thank the reviewers for the additional comments which help to further improve it.

Please update Figure 1C to include a y-axis and clarify that this figure reflects the strength of the correlation with predefined seed regions and not the number or proportion of voxels within a seed region with significant correlation (as in Figure 1B).

We added the corresponding axis title in Figure 1C (Functional connectivity) and clarified what Figure 1B and 1C represent in the corresponding Figure legend, as follows:

“Figure 1. Striatal functional connectivity is reduced in symptomatic HD mice. […] Data are represented as mean ± SEM (WT n=11 and R6/1 n=13 mice). *p<0.05, **p<0.01, ***p<0.001 HD *vs* WT.”